# PAX8 and MECOM are interaction partners driving ovarian cancer

Melusine Bleu[1,9], Fanny Mermet-Meillon[1,9], Verena Apfel[1,9], Louise Barys[1,9], Laura Holzer[1], Marianne Bachmann Salvy[1], Rui Lopes[1], Inês Amorim Monteiro Barbosa[1], Cecile Delmas[2], Alexandra Hinniger[2], Suzanne Chau[2], Markus Kaufmann[2], Simon Haenni[2], Karolin Berneiser[2,8], Maria Wahle[2], Ivana Moravec [3], Alexandra Vissières[3], Tania Poetsch[3], Erik Ahrné[3], Nathalie Carte[3], Johannes Voshol [3], Elisabeth Bechter[1], Jacques Hamon[1], Marco Meyerhofer[1], Dirk Erdmann[1], Matteo Fischer[1], Therese Stachyra[1], Felix Freuler[2], Sascha Gutmann [2], César Fernández[2], Tobias Schmelzle[1], Ulrike Naumann [2], Guglielmo Roma [2], Kate Lawrenson [4], Cristina Nieto-Oberhuber[5], Amanda Cobos-Correa[2], Stephane Ferretti[1], Dirk Schübeler[6,7] & Giorgio Giacomo Galli [1✉]

The transcription factor PAX8 is critical for the development of the thyroid and urogenital system. Comprehensive genomic screens furthermore indicate an additional oncogenic role for PAX8 in renal and ovarian cancers. While a plethora of PAX8-regulated genes in different contexts have been proposed, we still lack a mechanistic understanding of how PAX8 engages molecular complexes to drive disease-relevant oncogenic transcriptional programs. Here we show that protein isoforms originating from the *MECOM* locus form a complex with PAX8. These include MDS1-EVI1 (also called PRDM3) for which we map its interaction with PAX8 in vitro and in vivo. We show that PAX8 binds a large number of genomic sites and forms transcriptional hubs. At a subset of these, PAX8 together with PRDM3 regulates a specific gene expression module involved in adhesion and extracellular matrix. This gene module correlates with PAX8 and MECOM expression in large scale profiling of cell lines, patient-derived xenografts (PDXs) and clinical cases and stratifies gynecological cancer cases with worse prognosis. PRDM3 is amplified in ovarian cancers and we show that the MECOM locus and PAX8 sustain in vivo tumor growth, further supporting that the identified function of the MECOM locus underlies PAX8-driven oncogenic functions in ovarian cancer.

[1] Disease area Oncology, Novartis Institutes for Biomedical Research, Basel, Switzerland. [2] Chemical Biology and Therapeutics, Novartis Institutes for Biomedical Research, Basel, Switzerland. [3] Analytical Sciences and Imaging, Novartis Institutes for Biomedical Research, Basel, Switzerland. [4] Cedars-Sinai Women's Cancer Program at the Samuel Oschin Cancer Center, Los Angeles, CA, USA. [5] Global Discovery Chemistry, Novartis Institutes for Biomedical Research, Basel, Switzerland. [6] Friedrich Miescher Institute for Biomedical Research, University of Basel, Basel, Switzerland. [7] Faculty of Science, University of Basel, Basel, Switzerland. [8] Present address: Biozentrum, University of Basel, Basel, Switzerland. [9] These authors contributed equally: Melusine Bleu, Fanny Mermet-Meillon, Verena Apfel, Louise Barys. ✉email: giorgio.galli@novartis.com

Ovarian cancer is a heterogeneous disease accounting for >140,000 yearly deaths worldwide[1]. Despite the development of new treatment paradigms, the improvement of overall survival of ovarian cancer patients over the past decade has been dismal[2], highlighting the need to identify new therapeutic targets, particularly for subtypes not linked to specific genetic aberrations, such as *BRCA1/2* mutation.

Transcription factors (TFs) are key proteins governing lineage-specific gene expression programs[3]. Epigenomic profiling revealed how a subset of TFs in each cell type engage highly active regulatory elements to drive the expression of genes important for physiological or pathological cell states[4]. Large-scale functional genomic screens have identified critical TFs necessary for lineage-specific proliferation of cancer cells[5,6] and, in the case of ovarian cancer, indicated PAX8 as a key driver of cancer cell proliferation[6]. PAX8 is mostly known as a developmental TF required for the establishment of follicular thyroid cells in mice and humans[7,8]; however, its role in cancer is still under investigation. We and others have previously reported cell cycle and metabolism gene expression programs controlled by PAX8 in the kidney or ovarian cancer cells by binding to enhancer elements[9–11]. While a plethora of PAX8 target genes have been reported both in physiological and pathological contexts, a mechanistic understanding of how PAX8 exerts its oncogenic functions remains to be determined.

Here, we report the binary interaction between PAX8 and the products of the *MECOM* (MDS1-EVI1 complex locus) locus and dissect its function. *MECOM* is a transcriptional unit originally constituted by two main promoters (separated by 500 kb) driving the expression of the MDS1 and EVI1 proteins. However, a splicing event that occurs frequently in ovarian cancer and acute myeloid leukemia (AML) leads to the expression of the fusion protein MDS1–EVI1[12–14]. This protein has been previously defined as PRDM3 due to the presence of a PR domain of histone methyltransferases[12,14]. We demonstrate that the PAX8 DNA-binding domain engages a large number of genomic sites and, at a small subset of loci, recruits PRDM3 via its PR domain and an array of C2H2 zinc fingers. This complex regulates a defined gene expression module involved in cell adhesion and extracellular matrix formation. We demonstrate that both PAX8 and MECOM are critical TFs to sustain in vivo growth of ovarian tumors, likely by MECOM acting as a PAX8 cofactor mediating a subset of PAX8 oncogenic functions. Importantly, we define a PAX8–MECOM gene signature that characterizes patients of gynecological cancers with poor prognosis. Our molecular dissection analysis pinpoints a potential strategy to target the interaction of these oncogenic TFs.

## Results

**PAX8 and MECOM reside in the same complex.** PAX8 is a TF involved in lineage specification of the thyroid and urogenital tract. Genetic screens point to PAX8 as a candidate oncogene for ovarian[6] and kidney cancers[9] by regulating a gene expression program controlling cell cycle and metabolic genes[9–11]. While PAX8 has been shown to activate gene expression by recruiting acetyltransferases[15], an unbiased characterization of the TFs engaged by PAX8 to elicit its oncogenic program is lacking. To characterize PAX8 interaction network, we utilized the BioID system[16], in which the prokaryotic biotin ligase BirA is fused to a gene of interest, allowing to label proximally engaged proteins. In order to study PAX8, we inserted a BioID-HA cassette into the endogenous PAX8 locus in IGROV-1 ovarian cancer cells using CRISPR-Cas9 (Supplementary Figure 1A). Fusion of the BioID tag with a T2A-mCherry cassette allowed enrichment by FACS sorting of positive integrants as evidenced by the

emergence of the PAX8-BioID-HA fusion protein in bulk population analysis (Supplementary Figure 1B). Indeed, most of the derived clones from such enriched populations display expression of the PAX8-BioID-HA fusion at the expected molecular weight (Supplementary Figure 1C).

These cells were then used in a BioID experiment to identify the proximal interactome of PAX8 using streptavidin enrichment and quantitative mass spectrometry (MS) (Fig. 1A). Differential analysis revealed 106 proteins specifically enriched in cells labeled with biotin vs. control samples (*P* value < 0.01 and Log FC > 1), including PAX8 itself (Fig. 1A and Supplementary Data 1). Gene ontology analysis of the obtained hits demonstrates enrichment for proteins involved in transcription and DNA repair (Supplementary Figure 1D), compatible with the known nuclear roles for PAX8[10,11,17].

Further inspection of the list of hits revealed several histones as well as proteins involved in DNA damage and a diverse set of chromatin modifiers belonging to multiple complexes (Supplementary Figure 1E). Given the cell type-specific nature of PAX8 oncogenic phenotype, we focused on lineage-specific TFs enriched in our BioID-MS experiment. We were particularly intrigued by MECOM as this gene has been shown to be frequently amplified in ovarian cancer[18]. The two best characterized proteins encoded by the *MECOM* locus are EVI1 and MDS1–EVI1 (hereafter called PRDM3), which differ by the presence of an N-terminal domain PR/SET domain reported to bear histone methyltransferase activity[19] or mediate protein–protein interactions[20] with different expression patterns across tissues. Interestingly, expression analysis of The Cancer Genome Atlas (TCGA) dataset revealed a particular enrichment of *PRDM3* expression over *EVI1* in ovarian cancer (Supplementary Figure 1F).

Western blot analyses from two different IGROV-1 PAX8-BioID clones readily validated the proximity engagement of two MECOM splice variants EVI1 and PRDM3 by PAX8 (Fig. 1C). This was further confirmed by endogenous co-immunoprecipitation experiments between *MECOM* variants and PAX8 in two cell lines (Fig. 1D). In addition, due to the enriched expression of PRDM3 in ovarian cancer cells, we validated that PRDM3 alone can interact with PAX8 by co-immunoprecipitation of ectopically expressed PAX8 and PRDM3 in HEK293 cells (Fig. 1E), as well as by complementation assay using a cellular NanoBit assay (Fig. 1F). Collectively, our data indicate that PAX8 and MECOM splice variants, including the ovarian cancer-specific PRDM3, reside in the same protein complex.

**PAX8 DNA-binding domain engages PRDM3 in a binary interaction.** In order to dissect the molecular basis for the interaction between PAX8 and PRDM3, we generated a large set of PAX8 mutants using a mammalian in vitro transcription–translation (IVTT) system coupled to interaction analysis by NanoBit (Fig. 2A). PAX8 is composed of a N-terminal DNA-binding domain called Paired (PRD), a conserved octapeptide (OP), a truncated homeodomain (HD), and a C-terminal transactivation (TA) domain (Fig. 2A). Deletion of the PAX8 DNA-binding domain blunted the luciferase signal compared to deletion of PAX8 TA domain (Fig. 2B) while displaying similar protein expression levels (Supplementary Figure 2A). Importantly, point mutations abolishing PAX8 DNA-binding capacity do not significantly affect its interaction with PRDM3 (Fig. 2B), suggesting that structural elements within the DNA-binding domain are necessary for PRDM3 binding. In order to understand if the PAX8 DNA-binding domain was sufficient to interact with PRDM3, we probed each PAX8 domain in a minimal reconstituted IVTT system, again

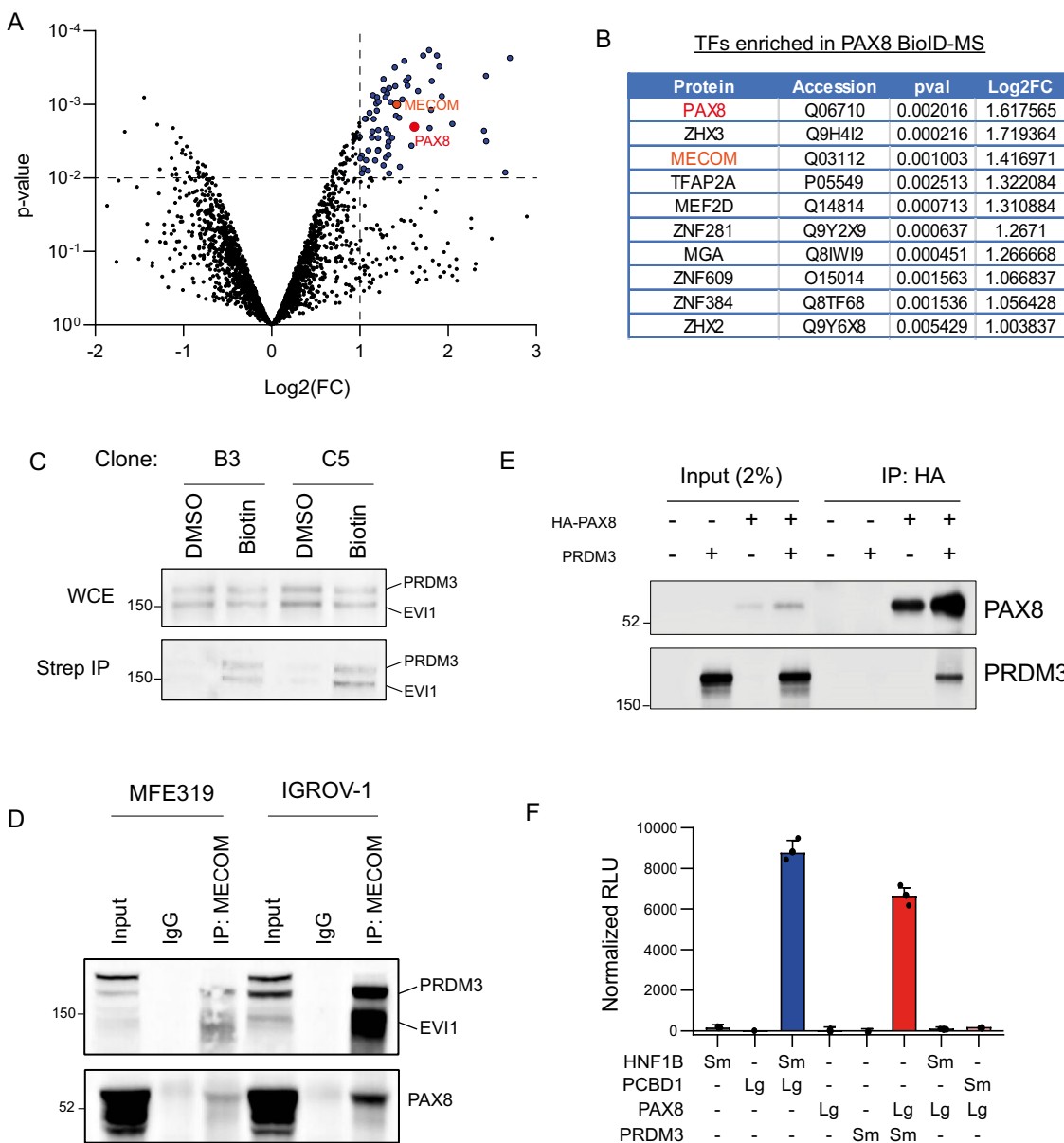

**Fig. 1 PRDM3 interacts with PAX8. A** BioID-MS results from IGROV-1-PAX8-BioID-T2A-mCherry cells. Blue dots represent proteins significantly enriched (*P* value <0.01 and Log FC >1). Red dot represents PAX8 and orange dot represents MECOM. **B** List of transcription factors enriched in PAX8-BioID IP-MS experiment. **C** Western blot from BioID-WB experiments in two different IGROV-1-PAX8-BioID-T2A-mCherry clones. The picture displays one representative image out of three independent experiments. **D** Endogenous co-immunoprecipitation between PAX8 and *MECOM* variants in MFE-319 and IGROV-1 cells. The picture displays one representative image out of three independent experiments. **E** Co-immunoprecipitation of ectopically expressed PAX8-HA and PRDM3 in HEK293 cells. The picture displays one representative image out of five independent experiments. **F** NanoBit assay in HEK293A cells transfected with PAX8-LgBit (Lg) and PRDM3-SmBit (Sm). HNF1B and PCBD1 are an unrelated pair used as positive and specificity controls. RLU relative luminescence unit. Data are presented as mean values ± SD from three biological replicates. Source data for Western blots and interaction measurements are provided as a Source Data file.

coupled to NanoBit (Supplementary Figure 2B). Only the PAX8 PRD domain gave a strong interaction signal when mixed to PRDM3, confirming that the PAX8 DNA-binding domain is sufficient for the interaction (Fig. 2C). Similarly, in order to understand which domains of PRDM3 are involved in PAX8 interaction, we expressed partially overlapping protein fragments scanning the entire length of PRDM3 with the IVTT-NanoBit system (Fig. 2D). While none of the constructs of single domains of PRDM3 were sufficient to achieve maximal interaction, we identified a construct encompassing the PR/SET domain plus the first array of zinc fingers (ZnF 1–7) to display optimal binding to PAX8 DNA-binding domain (Fig. 2E).

To test our results with an orthologous in vitro method, we expressed and purified recombinant proteins encompassing PAX8 PRD domain (amino acid residues 9–135) and PRDM3 PR/SET and ZnF 1–4 domains (amino acid residues 2–345) (Supplementary Figure 2C). Two-dimensional nuclear magnetic resonance (NMR) spectroscopy observing $^{13}$C-$^{15}$N-labeled PAX8 showed chemical shift perturbation for a subset of peaks upon addition of PRDM3 protein and substantial line broadening of all PAX8 resonances (Fig. 2F). In a complementary experiment where the methyl region of the proton spectra of PRDM3 was analyzed upon PAX8 addition, we observed peak broadening and changes in the resulting spectrum, significantly differing from the sum of the

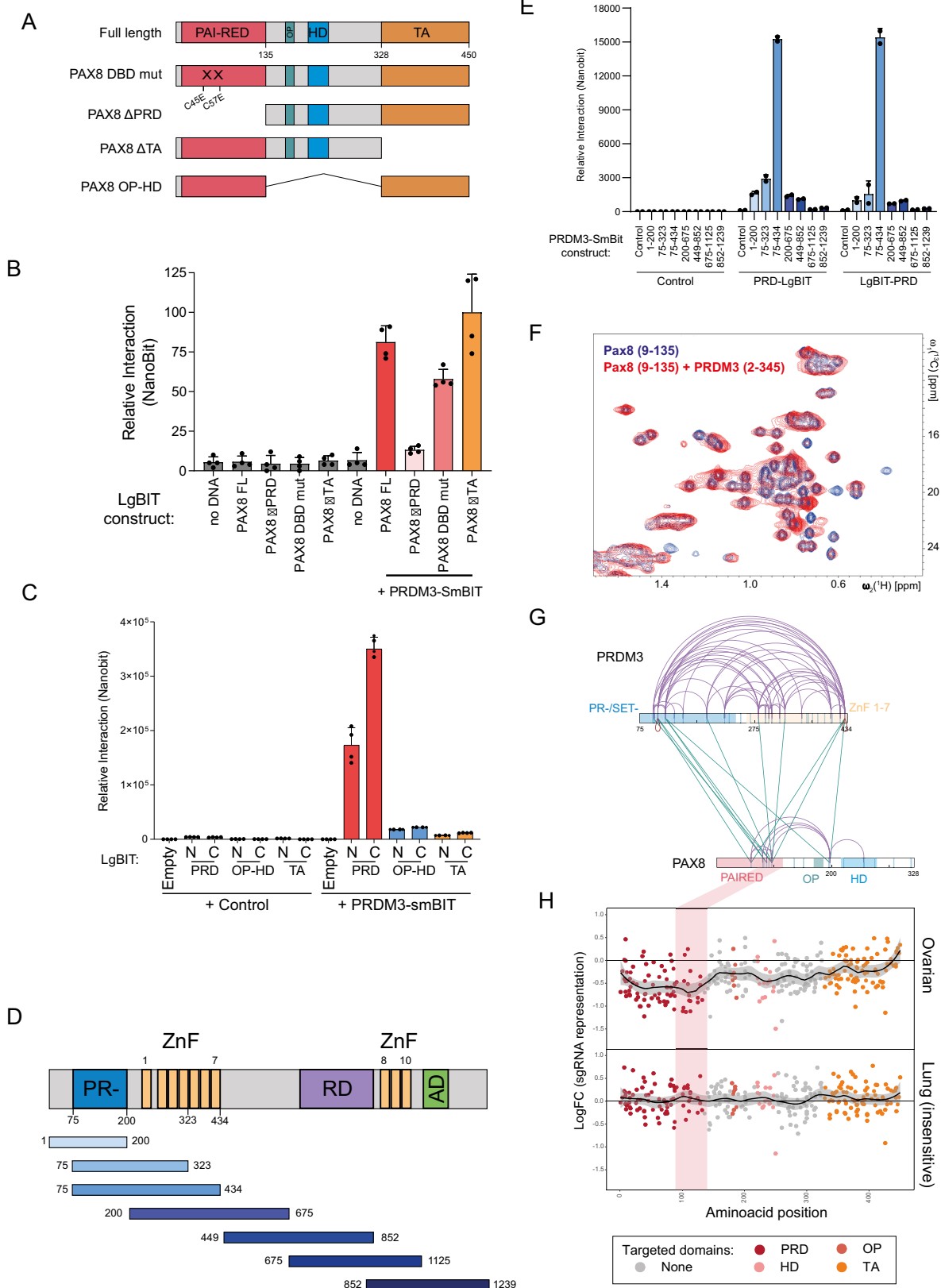

individual PRDM3 and PAX8 spectra (Supplementary Figure 2D). Together, this indicates interaction between the two proteins and argues for the formation of a binary complex independently of DNA presence. As a further test, we performed crosslinking MS between PAX8 and PRDM3. Treatment of a PAX8–PRDM3 complex with disuccinimidyl sulfoxide (DSSO) readily formed high-molecular-weight species observed by sodium dodecyl sulfate-polyacrylamide gel electrophoresis (SDS-PAGE) and matrix-assisted laser desorption ionization-time of flight MS (MALDI-MS) (Supplementary Figure S2E, S2F). Peptide mapping of these species revealed extensive intramolecular interactions between the PRDM3 PR domain and the neighboring ZnF array. In addition,

**Fig. 2 Mapping of the binary interaction between PAX8 and PRDM3. A** Schematic representation of PAX8 and corresponding mutants generated by in vitro transcription–translation (IVTT). **B** IVTT-NanoBit assay testing the interaction of PAX8-LgBit and truncation mutants with PRDM3-SmBit. Data are presented as mean ± SD from four biological replicates. **C** NanoBit assay testing the interaction of individual PAX8 domains with PRDM3-SmBit. N/C N-terminal and C-terminal tagging. Data are presented as mean ± SD from four biological replicates **D** Schematic representation of PRDM3 and corresponding protein fragments generated by IVTT. **E** NanoBit assay testing the interaction of PRDM3-SmBit protein fragments with the PAIRED domain of PAX8. LgBit-PRD/PRD-LgBit, N-terminal/C-terminal tagging. Data are presented as mean ± SD from two technical replicates from a representative experiment out of three independent experiments. **F** Overlay of the methyl region of 2D [$^{13}$C,$^{1}$H]-HMQC spectra of uniformly $^{13}$C,$^{15}$N-labeled PAX8(9–135) in the absence (blue) and in the presence of unlabeled PRDM3 (2–345) at equimolar concentration (red). **G** Crosslinking-MS results from PAX8 (2–328) and PRDM3 (75–434). Intramolecular interactions are marked in purple and intermolecular interactions in green. Vertical blue bars inside protein diagrams represent the position of lysine residues and shaded regions represent domain boundaries. **H** CRISPR-Tiling screen data in OV56 (ovarian) and NCI-H1299 (lung) cell lines. Each dot represents a single sgRNA, and color coding is based on targeting a specific domain in PAX8 protein. Shaded vertical bar represents region enriched in intramolecular crosslinks from (**G**) corresponding to second helical portion of PAX8 DBD (called –RED). Source data for Western blots and interaction measurements are provided as a Source Data file.

from both these regions, we observed intermolecular crosslinks converging onto the second half of the PAX8 PRD domain (Fig. 2G and Supplementary Figure S2G and Supplementary Data 2) further supporting the notion that both PR and ZnF 1–7 of PRDM3 are necessary to achieve optimal binding to PAX8. To validate our structural/interaction findings in cells, we performed a CRISPR-Tiling screening[21] in ovarian and lung cancer cells. Sliding window analysis of the single guide RNAs (sgRNA) representation displayed stronger dropout for sgRNAs targeting PAX8 DNA-binding domain (particularly the second half), compared to the other domains (Fig. 2H). Importantly, the phenotype is specific to ovarian cells, while lung cancer cells are inert to PAX8 targeting (Fig. 2H). Collectively, our data demonstrate that PAX8 and PRDM3 engage in a direct binary interaction involving PAX8 DNA-binding domain and the N-terminal portion of PRDM3.

**PAX8 recruits PRDM3 to chromatin.** In order to identify the functional consequences of the PAX8–PRDM3 interaction on gene regulation, we performed PAX8 and PRDM3 chromatin immunoprecipitation sequencing (ChIP-seq) (with an antibody raised against PRDM3 PR domain) in NIH:OVCAR3 ovarian cancer cells. This revealed that PAX8 locates in over 30,000 genomic regions, while we detected ~7600 PRDM3 sites (Fig. 3A). Importantly, the majority of these PRDM3 sites (>60%) were also bound by PAX8, arguing that both factors colocate on chromatin and further corroborating their tight relationship (Fig. 3A, Supplementary Figure S3A, and Supplementary Data 3). Both motif enrichment and de novo motif finding readily identified the PAX motif as significantly enriched in PAX8$^+$ and PAX8$^+$PRDM3$^+$ sites (Fig. 3B). At the same time, we did not detect evidence of the reported MECOM motif, suggesting that PRDM3 engages chromatin via PAX8 binding.

Next, we asked if chromatin binding of both factors occurs in a dependent or independent fashion. Towards this goal, we performed ChIP-seq for each factor upon downregulation of the other. RNA interference (RNAi) knockdown was validated by ChIP-quantitative real-time PCR (ChIP-qPCR) on selected loci (Supplementary Figure 3B). Subsequent differential binding analyses revealed that, upon PAX8 knockdown, PRDM3 displayed a global loss of occupancy (Fig. 3C, D) despite marginal changes of total protein levels (Supplementary Figure 3C). On the contrary, MECOM knockdown did not significantly affect PAX8 occupancy genome-wide (Fig. 3C, D). These results document a role for PAX8 in recruiting PRDM3/MECOM as a cofactor at common genomic sites.

Next, we analyzed the nuclear distribution of PAX8 and PRDM3 by transient expression of fluorescently tagged proteins and confocal microscopy. PAX8-eGFP was unevenly distributed in the nucleus and forming hub-like structures resembling transcriptional condensates[22] (Fig. 3E). In these hubs, ~40% of

PAX8 molecules are not mobile as evidenced by fluorescent recovery after photobleaching (FRAP). In contrast, PRDM3 displayed a more homogeneous nuclear expression and, at PAX8 hubs, rapidly diffused after photobleaching (Fig. 3E). This suggests strong tethering of PAX8 protein molecules in hub-like structures and a more dynamic engagement of PRDM3, potentially as a PAX8 cofactor.

Next, we wanted to define the transcriptional impact elicited by PAX8–MECOM complex and we performed RNA sequencing (RNA-seq) upon silencing of either *PAX8* or *MECOM* in a panel of five ovarian cancer cell lines, which display the highest sensitivity to both *PAX8* and *MECOM* knockdown (see below). Transcriptomic analyses were performed 4 days following short hairpin RNA (shRNA) induction using a doxycycline-inducible hairpin against *PAX8*[9] and two independent hairpins against *MECOM* displaying similar target knockdown efficiency (Supplementary Figure 3D and Supplementary Data 4). In order to identify common target genes of PAX8–MECOM, we regressed out potential cell line-specific effects, which identified a set of 58 genes that displayed significant changes upon silencing of *PAX8* or *MECOM* (Fig. 3F). Pathway enrichment analysis revealed that this *PAX8–MECOM* gene module was significantly enriched in genes functioning in the extracellular matrix, focal adhesions, and tumor growth factor-β signaling (Supplementary Figure 3E). Importantly, while either *PAX8* or *MECOM* depletion regulated this gene set consistently, the effect was stronger when depleting *PAX8* arguing that *MECOM* acts as a cofactor to modulate a subset of the *PAX8* target genes. In order to ask if this effect can be recapitulated in vivo and is not limited to cell lines in culture, we injected NIH:OVCAR3-shPAX8/shMECOM cells (two independent hairpins each) in nude mice and treated mice for 1 week with either vehicle or doxycycline for transcriptomic profiling (Supplementary Figure 3F). Importantly, also in vivo, our identified gene set was modulated by both *PAX8* and *MECOM* (Fig. 3F and Supplementary Data 4) and, again, shPAX8 perturbation inducing stronger transcriptional modulation, despite a milder knockdown efficiency (Fig. 3F and Supplementary Figure S3F). Collectively, our data suggest that PAX8 is a major TF in ovarian cancer cells by engaging a large number of genomic sites, while PRDM3 (MECOM) is specifically recruited by PAX8 at specific genomic loci to modulate a defined common gene module.

**PAX8 and PRDM3 drive ovarian tumor growth.** Large-scale functional genomic screens have classified PAX8 as an ovarian cancer dependency[6]. In light of our discovery of MECOM as an interactor of PAX8, we evaluated their relationship in such genetic screening datasets. We employed genome-wide RNAi or CRISPR datasets from DepMap (Dependency Map) and observed that, among ovarian models, the cell lines most sensitive to *PAX8*

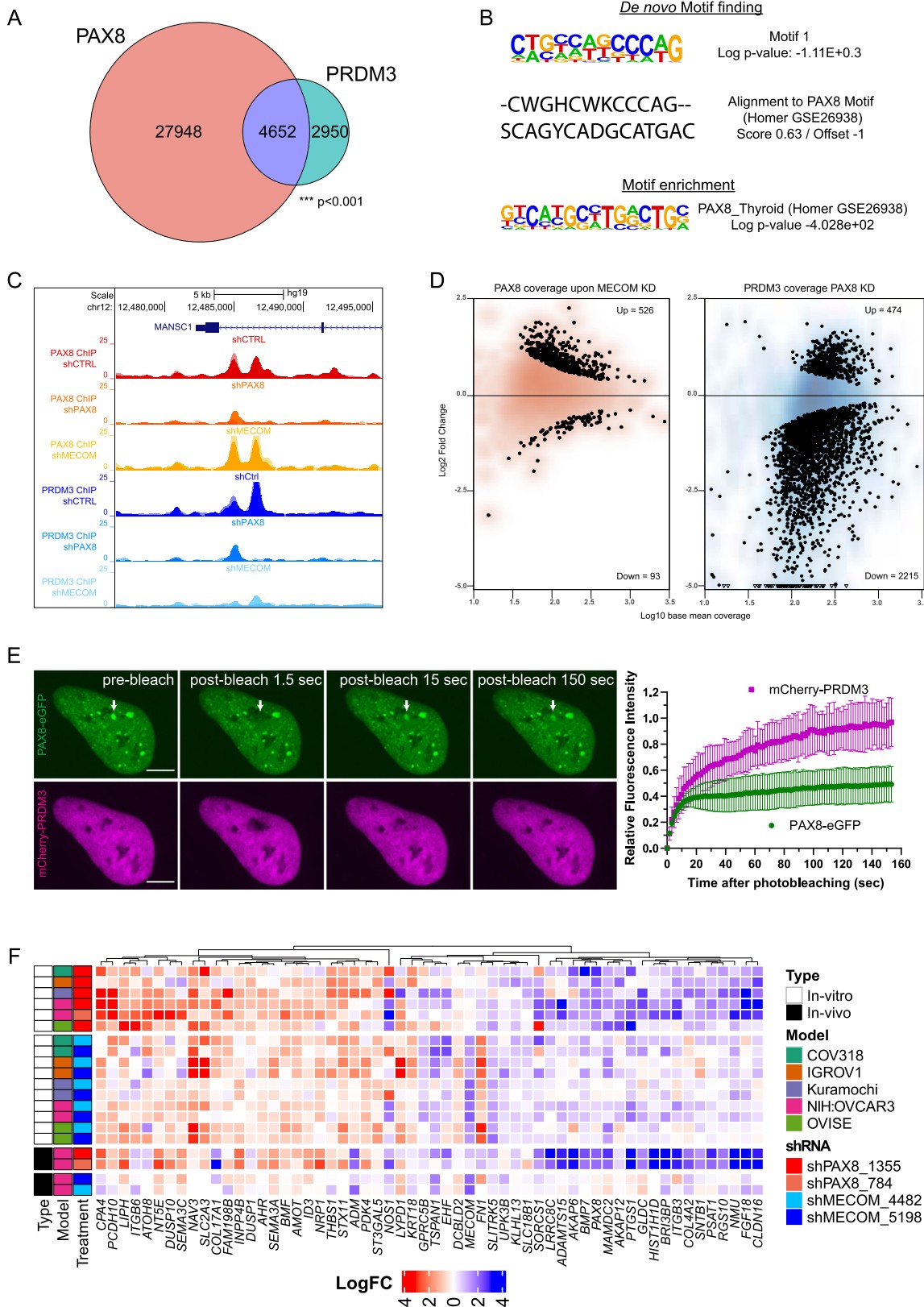

knowledge/knockout (KO) are also the ones most sensitive to *MECOM* perturbations (Fig. 4A and Supplementary Figure S4A). We rigorously tested these findings by performing colony formation assays in multiple cell lines with independent genetic reagents (Supplementary Figure 4B). A characteristic feature of *PAX8*-sensitive lines is high expression levels of *MECOM*, which

suggests the possibility that high *MECOM* expression could be a biomarker for *PAX8* sensitivity (Fig. 4A and Supplementary Figure S4A).

Next, we asked if *PAX8/MECOM* dependency can be recapitulated in xenograft models in vivo. Using NIH-OVCAR3 cells bearing doxycycline-inducible shRNAs against *PAX8* or

**Fig. 3 PAX8 recruits PRDM3 to common binding regions. A** Venn diagram showing the overlap of ChIP-seq peaks of PAX8 and PRDM3 in ovarian cancer cells. ***$P < 0.001$ represents the statistical significance of the overlap between PAX8 and PRDM3 using Fisher's exact test. **B** (Top) Sequence logo representation of the top motif identified by de novo motif finding in PAX8+PRDM3+ sites and alignment to known PAX8 motif. (Bottom) Motif enrichment analysis for known PAX8 motif in PAX8+PRDM3+ ChIP-seq peaks. **C** UCSC genome browser snapshot of the *MANSC1* locus showing ChIP-seq tracks of PAX8 and PRDM3 in ovarian cells following shRNA-mediated knockdown of PAX8 or MECOM. shCTRL is a negative control. **D** Differential binding analyses of PAX8 (left) and PRDM3 (right) upon MECOM or PAX8 knockdown, respectively. MA plot represents the distribution of Log FC (*y*-axis) and base mean coverage (*x*-axis). Dots represent peaks with statistically significant differences (numbers indicated). **E** Representative FRAP images of PAX8-eGFP (green) and mCherry-PRDM3 (magenta) signal in the nucleus of U2OS cell. Arrow points to the bleached region with PAX8 hub. Scale bar = 10 μm. Average FRAP curves and quantification were generated by EasyFRAP-web tool. Mobile fraction of PAX8 in bleached region = 0.6; half-recovery time $T_{1/2} = 7$ s; $R^2 = 1$. Mobile fraction of PRDM3 in bleached region = 1; half-recovery time $T_{1/2} = 15.2$ s; $R^2 = 1$. **F** Expression heatmap of 58 genes from gene modules identified from RNA-seq experiments in five ovarian cancer cell lines upon PAX8 or MECOM knockdown. Log FC for the same 58 genes from in vivo xenografts studies is also plotted. Source data for Western blots and qPCRs are provided as a Source Data file.

*MECOM* confirmed that *PAX8* silencing leads to profound regression (Fig. 4B), while *MECOM* loss induces tumor growth arrest/stasis (Fig. 4C). These different responses to *PAX8* or *MECOM* knockdown is suggestive of a weaker contribution of *MECOM* to ovarian cancer growth, possibly due to its cofactor activity. Importantly, upon long-term and potent ablation of *PAX8* in vivo, we observed a striking loss of MECOM proteins (Fig. 4D), while silencing of the latter left PAX8 levels unaffected (Supplementary Figure 4C). Such data are compatible with the role of PAX8 in recruiting PRDM3 to chromatin at a subset of common loci.

We then extended our findings to large-scale expression profiling of cell lines and patient-derived xenografts (PDXs). When models were ranked by the Signature score (*PAX8–MECOM* gene module derived from in vitro/in vivo RNA-seq in Fig. 3E), we observed a significant correlation with *PAX8* and *MECOM* expression (Supplementary Figure 4D, E), extending the notion that PAX8 and MECOM control of a specific signature is consistent across a large set of models. In addition, by binning TCGA high-grade serous ovarian cancer (HGSOC) cases ($n = 430$) based on either *PAX8* or *PRDM3* expression quartiles, we observed that cases with high *PAX8* and *MECOM* expression displayed significantly higher Signature Score compared to others (Fig. 4E). Survival analysis of ovarian and endometrial cancer patients displaying different levels of Signature Score revealed that cases displaying high Signature Scores (top quartile) exhibited significantly worse survival compared to patients with low scores (bottom quartile), suggesting that high PAX8/MECOM activity identifies a subset of patients bearing particularly aggressive tumors.

## Discussion

We here report a detailed mechanistic characterization of the interaction between PAX8 and the *MECOM* gene product PRDM3. Using cellular, biochemical, and biophysical methods, we map this binary interaction to the PAX8 DNA-binding domain and PRDM3 PR domain and ZnF 1–7. We demonstrate that PAX8 binds a large number of genomic sites owing to its hub-like nuclear distribution, while PRDM3 gets recruited to a subset of PAX8 sites to drive a gene expression module involved in the extracellular matrix and cell adhesion. Importantly PAX8 and MECOM are necessary for ovarian tumor growth in vivo and their signature distinguishes a subset of patients with poor prognosis.

PAX8 was originally identified as a critical TF for thyroid development. *PAX8* KO mice die due to the lack of follicular thyroid cells[8], while humans with mutations in PAX8 DNA-binding domain only develop a smaller but hyperactive thyroid[7], suggesting that anti-cancer therapies aiming at targeting PAX8 might bear a favorable therapeutic index. Importantly, the thyroid defects of *PAX8* KO mice can be rescued by administration of

levothyroxine revealing an infertility phenotype due to defects in the urogenital tract[23,24]. In line with PAX8 expression in fallopian tube secretory epithelial cells (FTSEC)[25], *PAX8* KO female mice display morphogenetic defects leading to a nonfunctional uterus. Importantly, when genetic lesions frequently occurring in ovarian cancer are induced in *PAX8*+ FTSEC, tumors resembling HGSOCs occur[25,26], suggesting that PAX8 is expressed in the cells of origin of HGSOC. While PAX8 is a key driver of ovarian development and carcinogenesis, transcriptomic, and epigenomic profiling highlighted substantial differences between the gene expression program of normal and cancerous FTSECs[10,11], indicating that PAX8 activity rather than expression differentiates the pathological state of ovarian cells.

PAX8 has been shown to interact with thyroid-specific TFs such as TTF-1 (also called NKX2-1)[27]. In ovarian cancers, PAX8 has been reported to interact with the Hippo pathway effectors YAP and TEAD[10]. While we do also observe in our PAX8 ChIP-seq dataset a strong enrichment for TEAD motifs, our study focuses on MECOM interaction as we readily identified it in BioID experiments and demonstrate that MECOM is necessary for in vivo tumor growth. It is thereby tempting to speculate that larger PAX8-containing complexes including several TFs could coexist in ovarian cancer cells, each potentially bearing locus and function specificity.

MECOM is of particular interest due to the complexity of its transcriptional unit. Originally classified as two separate units 500 kb apart encoding *MDS1* and *EVI1* genes, it was later re-annotated as a unified unit due to the expression of splice variants that include exons from both loci[13]. In particular, a splicing event from exon 2 of *MDS1* to exon 2 of *EVI1* generates a new protein (MDS1–EVI1) containing a PR domain, thereby defined also as PRDM3[12–14]. The PR domain shares 20–30% sequence similarity to the SET domain of histone methyltransferases, thereby bearing catalytic activity[12,14]. PRDM3 has been reported to be an H3K9me1 enzyme in the cytoplasm[19]; however, as we failed to detect intrinsic methyltransferase activity, we focused on the potential for the PR domain to be involved in protein–protein interactions[20]. Our biochemical and biophysical analyses suggest a large interface encompassing the PR domain and the first array of PRDM3 zinc fingers (typically involved in sequence-specific DNA binding[28]) embracing the second portion of PAX8 DNA-binding domain (called –RED). While such interaction resembles the one reported between the PRDM14 PR domain and MTGR1 helices[29], the involvement of ZnF 1–7 of PRDM3 corroborates the PAX8-dependent recruitment of PRDM3 in the absence of the MECOM motif. Further structural and biophysical studies will be needed to understand the three-dimensional interplay between PAX8, PRDM3, and nucleosomal DNA.

The functions of PRDM3 (MDS1-EVI) and its promoter activity have been extensively described in the hematopoietic system[30] and bone development[31]. In disease settings, while

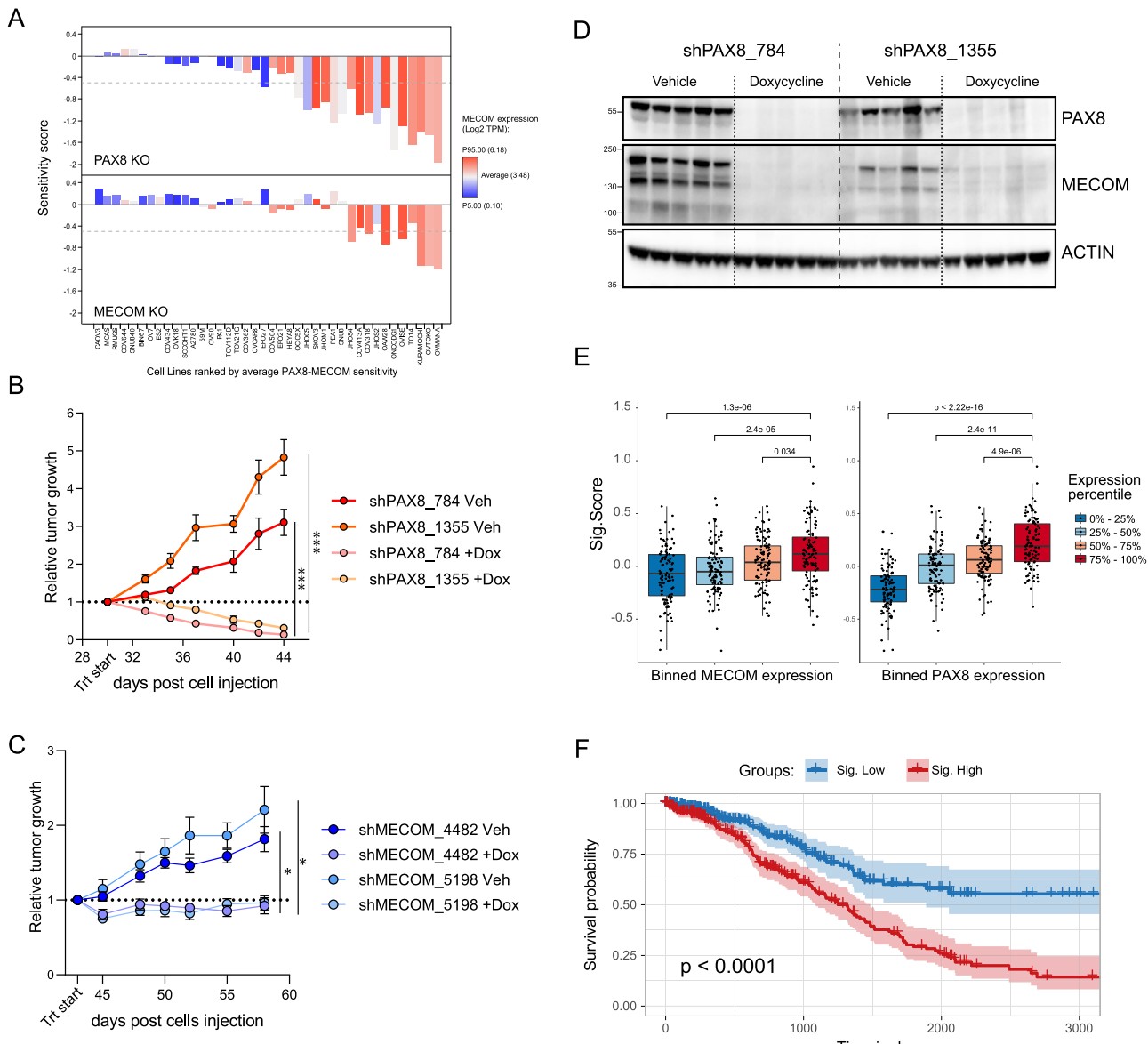

**Fig. 4 PAX8 and MECOM sustain ovarian cancer growth. A** Barplot showing sensitivity to PAX8 or MECOM KO as per CRISPR screens reported in DepMap portal. Bars are color coded by MECOM expression. **B**, **C** Tumor volume measurements of NIH:OVCAR3 cells bearing shRNAs against PAX8 (**B**) or MECOM (**C**). Trt start = day of starting daily doxycycline treatment. *$P < 0.01$ and ***$P < 0.0001$ signify significantly and highly significant differences to the respective vehicle (two-sided $t$ test post hoc) on the last treatment day. Data are presented as mean ± SEM from $n > 5$ mice cohorts. **D** Western blot analysis of tumors from (**B**) 1 week after treatment start. **E** Boxplot of $z$-score expression of PAX8–MECOM Signature (Sig. Score) in TCGA ovarian cases ($n = 608$) binned according to MECOM (left) or PAX8 (right) expression quartiles. Boxplots represent median and first and third quartiles, and whiskers extend to 95th percentile. $P$ values are based on two-sided Wilcoxon's rank-sum test. **F** Kaplan–Meier curve of survival from TCGA ovarian and endometrial patients bearing high or low levels of Signature Score (top and bottom quartile, $n = 746$). P $P$ value from log-rank test. Source data for Western blots and qPCRs are provided as a Source Data file.

PRDM3 has been reported as one of the driver oncogenes of AML[32,33], its role in solid tumors has never been explored in detail. We here report an additional role of PRDM3 in ovarian cancer as a PAX8 cofactor. In such a disease setting, the *MECOM* locus undergoes frequent amplification[18] and *PRDM3* expression (due to the locus alternative splicing event) is frequently occurring as evidenced by TCGA data analysis.

Together with PAX8, PRDM3 regulates a gene expression module involved in the extracellular matrix and cell adhesion. Recent single-cell RNA-seq analyses of FTSEC revealed cellular heterogeneity characterized by the expression of specific signatures[34]. Importantly, among the identified gene modules, the one related to EMT, which is expressed from *PAX8*+ cells, once

applied to the TCGA dataset, identifies a population of patients with poorer prognosis[34]. The fact that the genes positively regulated by PAX8 and MECOM also stratify patients with poorer prognosis suggests that indeed PAX8 cooperates with MECOM to regulate a gene expression program that promotes aggressive tumor phenotypes.

Combined, this study lays the foundation for studying PAX8 and MECOM as therapeutic targets for epithelial ovarian cancer. While attempts at identifying chemical matter inhibiting independently PAX2-5-8 or MECOM DNA-binding domains have been reported[35,36], identification of ligands inhibiting their functions remains particularly challenging due to the disordered/flexible nature of the proteins as well as the paucity of the

structure/function data. Our report sheds light on a previously uncharacterized interaction surface between two oncogenic TFs, potentially guiding the rational design of novel therapeutics for the treatment of a population of ovarian cancer patients with a poor prognosis.

## Methods

**Cell culture.** HEK293A and COV-318 were maintained in Dulbecco's modified Eagle's medium (DMEM) (Amimed) supplemented with 10% fetal bovine serum (FBS) (Seradigm), 1× L-glutamine (2 mM), 1× sodium pyruvate (1 mM), and 1× non-essential amino acid (0.1 mM). IGROV-1, Kuramochi, and Ovise were maintained in RPMI (Amimed) supplemented with 10% FBS, 1× L-glutamine, 1× sodium pyruvate, and 1× HEPES (10 mM). NIH:OVCAR3 were maintained in RPMI (Amimed) supplemented with 20% FBS, 1× L-glutamine, 1× sodium pyruvate, 1× HEPES, and 1× internal transcribed spacer. MFE-319 were cultured in RPMI:MEM (1:1), 20% FBS, 1× L-glutamine, 1× sodium pyruvate, 1× HEPES. All cell lines were obtained from ATCC and tested for identity by single-nucleotide polymorphism genotyping and mycoplasma contamination. Doxycycline-inducible shRNA cell lines were generated by lentiviral transduction of pLKO-TET-ON constructs containing the following shRNA sequences: shPAX8_1581 5′-gagagt-cacacaaaggaatct-3′; shMECOM_4482 5′-gaatgaacactccatagaaac-3′; and shME-COM_5198 5′-gcatgattcttctgattaaaa-3′. IGROV-1_Cas9 was obtained by lentiviral transduction of a construct overexpressing spCas9 under EF1A promoter (named pNGx-LV-c028). IGROV-1-Cas9-PAX8-BioID-HA-T2A-mCherry were generated by cotransfection of IGROV-1_Cas9 cells with the following sgRNAs (annealed synthetic crRNA with TracrRNA (IDT)) against exon 12 of PAX8 (PAX8_guide_1 5′-ctacagatggtcaaaggccg-3′ and PAX8_guide-2 5′-atggtcaaaggccgtggca-3′) and a repair template encompassing 800 bp upstream and downstream of the cleavage site flanking an in-frame cassette encoding a second-generation biotin protein ligase (BioID2) fused to mCherry sequence. DharmaFECT Duo Transfection Reagent was used. Positive signal cells were retrieved by FACS on a Sony Flow Cytometer model SH800S. Single-cell clones were then isolated and picked for further validation by Western blot.

**Gene expression analyses.** For Western blot analyses, cells were harvested and lysed in RIPA buffer supplemented with protease inhibitor cocktail (Roche). Protein samples were loaded on SDS-PAGE gels, transferred onto nitrocellulose membranes, and probed with the following antibodies: GAPDH (Cell Signaling, 8884; 1:1000 dilution), HA (BioLegend, 901501; 1:1000 dilution), LgBit (R&D systems, MAB10026, 1:1000 dilution), MECOM (Cell Signaling, 2593; 1:1000 dilution), PAX8 (Cell Signaling, 59019; 1:1000 dilution), PRDM3 (GenScript, U0869CG110-1; 1:10,000 dilution), VINCULIN (Sigma, V9131; 1:400 dilution), and HRP-anti-rabbit and HRP-anti-mouse (Cell Signaling).

RNA isolation was performed using QIAshredder (Qiagen) and RNeasy Plus Mini Kit (Qiagen) according to the manufacturer's recommendations. qRT-PCR was performed with QuantStudio 6 Flex (Applied Biosystems) using iTaq Universal Probes One-Step Kit (Bio-Rad) and the following Taqman probes: PAX8 (IDT, Hs.PT.58.1610472) and MECOM (IDT, Hs.PT.58.39825759). Gene expression levels were normalized to the HPRT housekeeping gene (Applied Biosystems). For ChIP-qPCR, qPCRs were performed using the Fast SYBR Green master mix reagent (Roche) with primers indicated in Supplementary Data 5.

### Interaction assays

*BioID-WB.* IGROV-1-Cas9-PAX8-BioID-T2A-mCherry Clone B3 and C5 cells were seeded at $4 \times 10^5$ cells/well in a 6-well plate. The following day, cells were treated with 50 μM biotin (Fluka) or dimethyl sulfoxide (Sigma) for 24 h. Cell pellets were harvested and lysed in RIPA buffer supplemented with protease inhibitor cocktail (Roche). Lysates (80 μg) were then incubated with streptavidin beads (Cell Signaling) overnight under rotation at 4 °C. Immunoprecipitates were washed five times with RIPA buffer containing protease inhibitors, eluted with NuPAGE Blue Sample buffer (Life Technologies) by incubation at 95 °C for 10 min, and resolved by standard SDS-PAGE gel electrophoresis and Western blotting as described above.

*Co-immunoprecipitation (co-IP).* For co-IP experiments, HEK293FT cells were transfected with HA-PAX8 FL and PRDM3 FL constructs using calcium phosphate, and lysed in NP40 buffer containing Protease Inhibitor Cocktail (Roche) 48 h after transfection. For endogenous co-IP experiments, IGROV-1 and MFE-319 cell pellets were collected and lysed in NP40 buffer supplemented with a protease inhibitor cocktail (Roche). Lysates (320 μg for HEK293FT; 2 mg of IGROV-1 and MFE-319) were pre-cleared using Protein A Agarose suspension beads (Millipore) for 1 h under rotation at 4 °C and then incubated with magnetic HA beads overnight under rotation at 4 °C (for HEK293FT) or with MECOM antibody, followed by incubation with Dynabeads Protein G (Invitrogen) for 1 h under rotation at 4 °C (for IGROV-1-1 and MFE-319). Immunoprecipitates were washed four times with NP40 buffer containing protease inhibitors, eluted with Nupage Blue Sample buffer (Life Technologies) resolved by standard SDS-PAGE gel electrophoresis, and Western blotting as described above.

*Cellular NanoBit assay.* HEK293A cells were plated at $1.25 \times 10^4$ cells/well in a white transparent bottom 96-well plate and transfected the following day with LgBiT-PAX8 FL, PRDM3 FL-SmBiT, SmBiT-HNF1β, PCBD1-LgBiT, and PCBD1-SmBiT. X-tremeGENE 9 (Roche) was used for transfection with a ratio 1:3 (DNA: transfection reagent). At 48 h post transfection, Nano-Glo cell reagent (Promega) was added and luminescence was measured using a Synergy HT reader (BioTek).

*In vitro transcription–translation.* PAX8-LgBit and PRDM3-SmBit constructs were synthesized by Twist Bioscience and cloned into a custom vector containing T7 promoter by T2S cloning. Constructs were expressed with the PURExpress In Vitro Protein Synthesis Kit (New England Biolabs [NEB #E6800S]) according to the manufacturer's protocol. Briefly, 250 ng plasmid DNA was added to a mixture of solutions A and B, and supplemented with a 20 U RNase inhibitor (Promega, #N2111). The reaction mixture was incubated for 2 h at 37 °C. After completion of the synthesis reactions, PAX8-LgBit and PRDM3-Smbit constructs were mixed with a 1:1 ratio in 25 μl of phosphate-buffered saline (PBS) in a white 96-well plate. For control conditions, PAX8-LgBit and PRDM3-Smbit constructs were mixed with PBS alone. After incubation for 20 min, at room temperature, 25 μl of Nano-Glo® Luciferase Assay Reagent (Promega) was added into each well. After a 5 min incubation at room temperature, relative luminescence units were measured using an Infinite 200 PRO TECAN Reader. In parallel, 2 μl of each reaction was mixed with NuPAGE Blue Sample buffer (Life Technologies) before loading the samples on SDS-PAGE gels for Western blotting as described above.

**CRISPR-Tiling.** OV56 and NCI-H1299 cells were engineered to express spCas9 and subsequently infected at a multiplicity of infection of 0.4 with a previously reported CRISPR-Tiling library[37] at 1000× representation. Cells were selected for 4 days in the presence of puromycin, and a reference sample was collected 72 h after selection to ensure adequate selection/representation. Cells were propagated for a total of 14 days with an average shRNA/sgRNA representation of ≥1000 maintained at each passage. Cells (100 million) were harvested for DNA extraction using the Qiagen QIAmp Blood Maxi Kit, shRNA and sgRNAs were PCR amplified from 100 μg of genomic DNA, and PCR fragments of 260–280 bp were purified using Agencourt AMpure XP beads (Beckman). The resulting fragments were sequenced on a HiSeq 2500 (Illumina) with a single end 50 bp run. Sequencing reads were aligned to the sgRNA library and, for each sample, the total number of read counts was normalized to $50 \times 10^6$, with five additional pseudo-counts added to each sgRNA to minimize false positives in the low-abundance tail of the sgRNA library distribution, where counts are unreliable. All samples had day 14 log 2 ratios for each sgRNA calculated relative to plasmid counts.

**Recombinant proteins expression and purification.** The DNA fragment encoding PAX8(9–135)-LgBit was PCR-amplified with primers comprising *Lgu*I restriction sites and cloned by Golden Gate into a pET-derived vector with an N-terminal His6-ZZ-Gly·Ser spacer-HRV3C affinity purification and solubilizing tag. The DNA fragment encoding PRDM3 (75–434)-SmBit was PCR-amplified with primers comprising *Lgu*I restriction sites for cloning by Golden Gate into a pET-derived vector with an N-terminal His6-StreptagII-spacer-Rbx-Gly·Ser spacer-HRV3C affinity purification and solubilizing tag. The DNA sequence of all expression constructs was verified by Sanger sequencing. The expression plasmids were transformed into BL21 (DE3)-competent *Escherichia coli* cells (New England Biolabs, Ipswich, MA) and grown overnight at 37 °C. LB medium was inoculated with a bacterial pre-culture and incubated under constant shaking at 37 °C. At OD600 = 0.8, the culture was chilled to 18 °C, and protein expression was induced by the addition of 1 mM isopropyl β-D-1-thiogalactopyranoside and run overnight. Bacterial cells were harvested by centrifugation at $4000 \times g$ for 20 min, frozen on dry ice, and stored at −80 °C. Recombinant PAX8(9–135)-LgBit and PRDM3 (75–434)-SmBit proteins were purified using different protocols.

*PAX8(9–135)-LgBit purification.* Cell pellets were thawed and suspended in buffer A (50 mM $NaH_2PO_4$, 0.3 M NaCl, 30 mM imidazole, pH 7.8) supplemented with cOmplete protease inhibitor (Roche, Switzerland) and TurboNuclease (Merck, Germany). The cells were mechanically disrupted by three passages through an EmulsiFlex C3 homogenizer (Avestin, Canada), and insoluble cell debris was removed by centrifugation for 30 min at 40,000 ×g. The clarified cell lysate was loaded onto 5 ml HisTrap HP columns (GE Healthcare, UK) mounted on an ÄKTA Pure chromatography system (GE Healthcare). Contaminating proteins were washed away with 10 column volumes of buffer A, and the His-tagged protein was eluted with a linear gradient over 10 column volumes to 100% buffer B (buffer A with 300 mM imidazole). The N-terminal purification tag was cleaved off overnight at 5 °C by GST-tagged HRV3C protease during dialysis against 20 mM $NaH_2PO_4$, 150 mM NaCl, 0.5 mM TCEP (tris(2-carboxyethyl)phosphine)), and 10% glycerol, pH 7.5. The dialyzed protein was diluted with an equal volume of 20 mM $NaH_2PO_4$, 10% glycerol, pH 7.5, before being loaded onto a Mono S 10/100 GL column (GE Healthcare) to remove the cleaved tag, HRV3C protease, and contaminating host cell proteins. The cleaved protein was eluted with a linear gradient of 20 mM $NaH_2PO_4$, 1 M NaCl, 0.5 mM TCEP, and 10% glycerol, pH 7.5. The fractions containing the PAX8(9–135)-LgBit proteins were pooled, concentrated with Amicon Ultra-15 10 K centrifugal filter unit (Merck, Germany), and loaded onto a HiLoad Superdex 75 16/600 pg size exclusion column (GE

Healthcare, UK) equilibrated with 20 mM HEPES, 150 mM NaCl, 0.5 mM TCEP, and 10% glycerol, pH 7.2. The fractions containing pure protein were pooled and concentrated to ~3 mg/ml in an Amicon filter unit (Merck, Germany). The purity and concentration of the protein samples were determined by reverse-phase ultra-high-performance liquid chromatography (RP-UHPLC), measuring the absorbance at 210 nm. The concentration was calculated using a bovine serum albumin (BSA) standard curve as a reference. Identity and molecular weight of the PAX8(9–135)-LgBit protein was confirmed by LC-MS.

*PRDM3 (75–434)-SmBit purification.* Cell pellets were thawed and suspended in buffer A (50 mM Tris, 300 mM NaCl, 10% glycerol, pH 8) supplemented with cOmplete protease inhibitor (Roche, Switzerland) and TurboNuclease (Merck, Germany). The cells were disrupted by sonication for 5 ×1 min at level 5 and amplitude 50% (Branson sonifier), and insoluble cell debris was removed by centrifugation for 30 min at 40,000 × g. The clarified cell lysate was loaded onto two 1 ml HisTALON columns (GE Healthcare, UK) mounted in series on an ÄKTA Pure chromatography system (GE Healthcare). Contaminating proteins were washed away with 10 column volumes of buffer A, and the His-tagged protein was eluted with a linear gradient over 10 column volumes to 100% buffer B (buffer A with 200 mM imidazole). The N-terminal purification tag was cleaved off overnight at 5 °C by His-MBP-3C protease during dialysis against buffer A. The cleaved protein was passed over the re-equilibrated HisTALON columns to remove the cleaved tag, HRV3C protease, and contaminating host cell proteins. The fractions containing PRDM3(75–434)-SmBit proteins were pooled, and protein concentration was ~0.05 mg/ml. The purity and concentration of the protein samples were determined by RP-UHPLC, measuring the absorbance at 210 nm. The concentration was calculated using a BSA standard curve as a reference. Identity and molecular weight of the PRDM3 (75–434)-SmBit protein was confirmed by LC-MS.

## Mass spectrometry

*BioID and quantitative MS.* For the BioID experiment, IGROV-1-Cas9-PAX8-BioID-T2A-mCherry cells were kept in (biotin-free) DMEM medium for 24 h to reduce endogenous biotinylation levels. Subsequently, cells were incubated with or without 10 µM biotin for 24 h and harvested for proteomic analysis. Cell pellets of biological triplicates for both conditions were lysed by sonication in RIPA lysis buffer (Millipore, 20–188 plus 0.1% SDS) and cleared by centrifugation for 20 min at 10,000 × g. Supernatants containing 3 mg of protein were incubated with 30 µl of streptavidin-agarose beads (Pierce High capacity Streptavidin-Agarose Resin) for 3 h at 4 °C. Beads were washed four times with RIPA buffer and transferred to 100 mM TEAB (triethylammonium bicarbonate) buffer (pH 8.5) containing 7 µg of a Trypsin/LysC mixture (Promega). On-bead digestion was carried out overnight at 37 °C, followed by the addition of another 2 µg of the enzyme mixture and further 1 h incubation. Subsequently, samples were centrifuged for 5 min at 300 × g to remove the beads and the supernatants were acidified to 1% trifluoroacetic acid (TFA) to stop the digestion. The resulting peptide mixtures were desalted using the PreOmics iST Kit according to the manufacturer's instructions and dried before tandem mass tag (TMT) labeling and liquid chromatography with tandem MS (LC-MS-MS). TMT 6-plex (Thermo Scientific) labeling was performed as per the manufacturer's instructions using 0.4 mg of TMT labeling reagent per sample. After checking the labeling efficiency (94%) by LC-MS-MS as below, the six samples were mixed and fractionated using the Pierce High pH Fractionation Kit (Thermo Scientific) following the manufacturer's protocol for TMT-labeled samples. Fractions were dried and analyzed by LC-MS-MS on a Q-Exactive HF-X mass spectrometer equipped with an Easy-nanoLC (nLC) system (Thermo Scientific). Samples were injected onto an Easyspray PepMap RSLC C18, 2 µm, 100 A, 75 µm × 15 cm column equipped with a 2 cm trap column (Thermo Scientific) and eluted with a 90 min gradient from 2 to 32% acetonitrile in 0.1 formic acid. Mass spectrometric analysis was performed using a top 15 data-dependent acquisition method using the following settings: MS1 resolution 60,000 with a maximum injection time of 50 ms and an AGC target of 1E6 ions; MS-MS resolution 30,000 with a maximum injection time of 40 ms and an AGC target of 1E5 ions; and a stepped normalized collision energy setting of 27, 30, and 33. Raw files were processed with Proteome Discoverer Software (Thermo Scientific, version 2.1) and in-house Python scripts for statistical analysis. Functional annotations were performed using DAVID (https://david.ncifcrf.gov/) using the Gene Ontology Biological Process All annotation.

*Crosslinking MS.* Crosslinking MS was carried out essentially as described before[38]. Briefly, PAX8 (2–328, AviTag) and PRDM3 proteins were mixed in a 1:1 ratio (7 µM) each, preincubated for 1 h at room temperature, and crosslinked with 1 or 2 mM DSSO (Thermo Scientific) for 1.5 h at room temperature and quenched with 20 mM NH₄HCO₃. The degree of covalent complex formation was evaluated by SDS-PAGE and MALDI-MS (Ultraflextreme II, Bruker), using the dried droplet method with a saturated sinapinic acid solution in CH₃CN/H₂O at a ratio of (75:25; v:v) with 0.1% TFA (v:v). MALDI-MS analyses were performed in linear mode using an external calibration with the protein calibration standard II (Bruker). For peptide level analysis, the crosslinked complex was processed using the PreOmics iST Kit (PreOmics) according to the manufacturer's instructions. The final peptide samples were dried and resuspended in 2% acetonitrile in 0.1% formic acid for LC-MS-MS analysis using a Lumos Fusion mass spectrometer, equipped with an Easy-nLC 1200 (Thermo Scientific). RP chromatography was performed on an Easyspray PepMap RSLC C18,

2 µm, 100 A, 75 µm × 15 cm column (Thermo Scientific). Crosslinked peptides were separated with a 180 min gradient from 2 to 80% of acetonitrile in 0.1% formic acid at a flow rate of 300 nl/min. MS data were acquired using the previously described CID-MS2-MS3 method[39]. LC-MS/MS data were analyzed with the default Proteome Discoverer (versions 2.2–2.4, Thermo Scientific) scripts using the XlinkX node against a fasta database containing the two proteins of interest. Only crosslinked peptides with an XlinkX score >50 are reported. The protein–protein interaction mapping for the complex was visualized with the xiNET viewer tool[40].

**Nuclear magnetic resonance**. Solutions of uniformly ¹³C,¹⁵N-labeled PAX8(9–135) and unlabeled PRDM3(2–345) were prepared at a concentration of 100 µM in NMR buffer (25 mM d₁₁-Tris, 100 mM NaCl, 1 mM d₁₆-TCEP, 10% D₂O, pH = 8.0). All NMR experiments were performed at protein concentrations of 50 µM, which were achieved either by carefully mixing both protein solutions at a 1:1 ratio for the complex or by diluting the individual proteins with the NMR buffer for the reference spectra. All samples contained 200 µM of 2,2-dimethyl-2-silapentane-5-sulfonate-d₆ sodium salt, which was used as an internal standard. The NMR spectra were measured in 3 mm NMR tubes with a sample volume of 180 µl. For each sample, 1D ¹H and 2D [¹³C,¹H]-heteronuclear multiple quantum coherence spectra were recorded at 296 K on a Bruker Avance III HD 800 MHz NMR spectrometer equipped with a 5 mm ¹H {¹³C,¹⁵N}-triple resonance cryogenic probe with shielded xyz-gradient coils. The data were processed and analyzed with the software Topspin 3.6 (Bruker, Switzerland).

## NGS-based technologies

*RNA sequencing.* RNA was prepared from cells or tumors using RNeasy Mini Kit (Qiagen) and RNA-seq libraries were prepared using TruSeq RNA Library Prep Kit v2 (Illumina) according to the manufacturer's recommendations. Libraries were sequenced on a HiSeq 2500 (Illumina).

*ChIP-sequencing.* Cells were crosslinked in 1% formaldehyde in PBS for 10 min at room temperature, after which the reaction was stopped by the addition of 0.125 M glycine. Cells were lysed and harvested in ChIP buffer (100 mM Tris at pH 8.6, 0.3% SDS, 1.7% Triton X-100, and 5 mM EDTA) and the chromatin disrupted by sonication using a EpiShear sonicator (Active Motif) to obtain fragments of average 200–500 bp in size. One hundred micrograms of chromatin was incubated with specific antibodies overnight. Antibodies used are PAX8 (Cell Signaling, 59019, 1:10); PRDM3 (GenScript, U0869CG110-1, 1.5 µg). Immunoprecipitated complexes were recovered on Protein G Dynabeads (Invitrogen), DNA was recovered by reverse crosslinking (65 °C for 8 h in 1% SDS and 0.1 M NaHCO₃ in TE buffer), and purified using SPRI Select beads (Beckman Coulter). Libraries for ChIP-seq were generated using Ovation® Ultralow Library System V2 (NuGEN) and barcodes were added using NEBNext Multiplex Oligos for Illumina (Index Primers Set 1) (NEB) according to the manufacturer's recommendation.

**Confocal imaging and FRAP**. U2OS cells were seeded on 8-well chambered ibi-Treat coverslip (ibidi, #80826) at the density of 10,000 cells/well in DMEM + 10% FBS (no antibiotics). After 2 days of incubation, cells were transiently co-transfected with constructs expressing eGFP/mTurquoise2 fusions of PAX8 and mCherry/mVenus fusions of PRDM3 (125 ng:125 ng of DNAs/well) by Lipofectamine 3000 (Invitrogen, #L3000001) according to the manufacturer's procedure. Images were acquired 1 day after transfection by Olympus confocal laser scanning microscope (FV3000) using UPLSAPO ×60 S2 silicon oil immersion lens (NA 1.3) at 37 °C. Fluorescent proteins were excited by corresponding low intensities of laser lines 488 nm (eGFP), 561 nm (mCherry), 445 nm (mTurquoise2), and 514 nm (mVenus). Fluorescence emission was collected between 500 and 540, 570 and 620, 450 and 500, and 530 and 620 nm, respectively.

To measure the baseline fluorescence intensity before bleaching the selected region of interest, 5 frames pre-bleaching were recorded, followed by bleaching for 0.5 s with 488 nm laser of high power without scanning, and recording for the next 95 frames post bleaching. The time interval between scanned frames was 1.5 s. The shape of the bleached region was circular with a diameter of 20 pixels (=~1.9 µm). Average fluorescence intensities in the region of photobleaching, in the whole area of the nucleus, and in a background region outside the nucleus were monitored over time. The position of the observed bleached region was manually adjusted for cell movement over the period of observation. Five regions of photobleaching (each in an individual cell) were monitored in total within a single experiment. Displayed average FRAP curves were plotted by GraphPad Prism version 8.4.2 for Windows (GraphPad Software, San Diego, CA, USA). Nonlinear fitting of the average FRAP curves and quantification were generated by an open tool EasyFRAP-web[41].

**Animal experimentation**. All animal experiments were performed according to procedures covered by permit number BS-1975 issued by the Cantonal Veterinary Office, Basel, Switzerland, and strictly adhered to the federal animal protection act and the federal animal protection code. All animals were permitted to adapt for 7 days and housed in a pathogen-controlled environment (five mice/type III cage) with access to food and water ad libitum and were identified with radio frequency identification transponders.

NIH:OVCAR3 bearing different doxycycline-inducible shRNAs (10 million cells in Hank's balanced salt solution:Matrigel 1:1) were subcutaneously injected in the

flank of 6–8-week-old female athymic nude mice (Charles River). When tumors reached a mean tumor volume of ~100–150 mm$^3$, animals were randomized into different treatment groups based on similar tumor size ($n > 6$/group). Tumor size was measured three times a week with a caliper. Tumor volume was calculated using the formula (length × width$^2$) × $\pi$/6 and expressed in mm$^3$. Data are presented as mean ± SEM. Differences between the changes in TVol were assessed on the endpoint using a $t$ test post hoc. At completion of the experiment, mice were euthanized according to the protocol, tumors were isolated, snap frozen in liquid nitrogen, and pulverized for molecular analyses using Covaris CP02.

### Bioinformatic analyses

*RNA sequencing.* Gene-level expression quantities were estimated by the Salmon algorithm[42]. Differential expression analysis was performed with DESeq2[43]. PAX8–PRDM3 gene signature was identified by selecting the median top and bottom 29 genes significantly modulated genes upon knockdown of PAX8 or MECOM across all models tested in vitro. Raw data are currently being deposited to SRA.

*ChIP-sequencing.* ChIP-seq data were mapped to the *human* reference genome (hg19 assembly) using bowtie2[44]. Duplicated reads were removed using the MarkDuplicates utility of the Picard tools (http://broadinstitute.github.io/picard) and peaks called using macs2[45] version 2.1.1 using a P value cutoff of -p 0.000000001. ENCODE blacklisted regions[44] were dismissed and an IDR[45] (irreproducible discovery rate) cutoff of 0.05 was applied on sample replicates. For global analyses of PAX8–PRDM3 binding, the union of PAX8 and PRDM3 peaks was used, whereas the overlap for co-occupied peaks was identified using DiffBind[46] R package. The significance of overlaps between PAX8 and PRDM3 binding sites was assessed using a permutation test from R package ChIPpeakAnno[46] (peakPermTest, version 3.24.1). The consensus PAX8 peaks were centered on the cell line-specific peak summits. The ChIP-seq dataset was then plotted over a 5 kb window centered at summits and organized according to the three clusters. Heat maps were generated with the genomation R/Bioconductor package interface[47]. Motif finding was performed using Homer (http://homer.ucsd.edu/homer/motif/) with default parameters. Differential binding at the common PAX8–PRDM3 sites was performed on extracted binding read counts matrices using DESeq2[43]. Raw data are currently being deposited to SRA.

*Intersection with publicly available datasets.* Normalized Z-scores for expression of the 58 genes PAX8–PRDM3 modules were calculated for CCLE[48], PTX[49], and TCGA (https://portal.gdc.cancer.gov/) datasets for samples belonging to ovarian lineage and compiled in a signature score (the sign of genes identified by RNA-seq as inversely regulated by shPAX8/shMECOM were reversed). Correlations between PAX8 or MECOM expression and median signature score were performed according to the Pearson method. TCGA cases were divided into four equal-sized bins based on either MECOM or PAX8 expression and differences in Signature Score were analyzed by Wilcoxon's test. For Kaplan–Meier curves, ovarian (ovarian serous carcinoma) and endometrial (uterine corpus endometrial carcinoma) cases were divided into the top or bottom quartile of expression of genes directly regulated by PAX8–MECOM and survival curves were generated in R using the survival https://cran.r-project.org/web/packages/survival/index.html and survminer https://cran.r-project.org/web/packages/survminer/index.html package.

**Reporting summary**. Further information on research design is available in the Nature Research Reporting Summary linked to this article.

## Data availability

The Crosslinking_MS data used in this study are available in the PRIDE database under accession code PXD021708. The BioID-MS data generated in this study have been deposited in the PRIDE database under accession code PXD021709. ChIP-seq and RNA-seq data have been deposited in SRA under accession PRJNA655844 and PRJNA655836, respectively. The remaining data are available within the Article, Supplementary information, or available from the authors upon request. Source data are provided with this paper.

## Code availability

Computational analyses have been performed using open source code as indicated in the "Methods" section. No proprietary code/softwares have been employed.

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

## Acknowledgements
We thank David Ruddy, Michelle Piquet, and Marc Altorfer for additional technical support and members of the Schübeler lab for insightful discussion.

## Author contributions
M.B., V.A., F.M.-M., R.L., I.B., M.K., and A.C.-C. performed molecular and cellular biology experiments. L.B. and M.B.S. performed bioinformatic analyses. L.H. and S.F. performed and supervised the in vivo experiments. C.D., A.H., S.C., S.H., K.B., M.W., E.B., J.H., M.M., D.E., M.F., T.St., F.F., S.G., C.F., and C.N.-O. performed or supervised biochemical and biophysical measurements. I.M. performed imaging experiments. A.V., T.P., E.A., N.C., and J.V. performed or supervised mass spectrometry experiments. U.N. and G.R. supported NGS measurements. G.G.G. supervised the project and wrote the manuscript together with insights by T.S.c., K.L., and D.S.

## Competing interests
All the authors affiliated with Novartis Institutes for Biomedical Research are employees of Novartis. The remaining authors declare no competing interests.
