## [Peer Review File · Nature Communications]

REVIEWER COMMENTS

Reviewer #1 (Remarks to the Author):

This is a very comprehensive study of the role of PAX8 and MECOM in HGSOV. While PAX8 and MECOM have been implicated in HGSOV development for some time, this study demonstrates a physical interaction/dependency and provides insights into potential mechanisms of how this drives HGSOV. The authors have been very thorough with their experimentation by providing orthogonal validations where possible. A strength of the study is the use and validation in multiple ovarian cancer cell lines. The manuscript would have been strengthened if in addition to knocking down PAX8 and MECOM expression in cancer cell lines, analyses could have been conducted in fallopian tumour epithelial, cell lines with induced expression. This could be an important experiment to understand the earliest events in HGSOV initiation since it seems PAX8 and MECOM are important early drivers of HGSOV.

The manuscript could probably be shortened as parts of the results include text which is more appropriate for the discussion (e.g. the last paragraph of the results). Also the discussion summarises a lot of the biology of PAX8 from thyroid cancer but I think it would be useful to include some discussion of other work that has proposed targeting PAX8/MECOM. Specifically, how does the results of this study aid in developing new therapeutics?

Overall the manuscript is well written, although there are a few sentences where the grammar is a bit awkward.

Results, page 3 "...its transcriptional network we utilized of the BioID system.

Results page 3 "...gene of interest allowing to label proximally"

Results, page 4 "while displaying similar protein expression levels confirmed (Figure S2A)".

Results page 6. "Importantly, lines highly sensitive PAX8 top were also highly sensitive to MECOM knockdown".

Discussion page 8. "PAX8 knockout (KO) mice die to the lack of formation of follicular".

Reviewer #2 (Remarks to the Author):

Bleu et al. describe a new interaction between a proto-oncogene originating the MECOM locus (MDS-EVI or PRDM3) and the transcription factor, PAX8. The PAX8-PRDM3 complex regulates genes involved in adhesion and extracellular matrix deposition and controls tumor cell growth. They further show that PRDM3 is amplified in ovarian cancer, and PAX8-PRDM3 regulated genes stratify ovarian cancer cases with poor prognoses. The role of PAX8 and expression of MECOM gene products in ovarian cancer have been reported and MECOM is known to play roles in other cancers. Although the authors provide strong biochemical data supporting a PAX8-PRDM3 interaction and demonstrate its physiological relevance in ovarian cancer, the functional consequences of the interaction are not well described in the data presented. How PRDM3 mediates PAX8 oncogenic transcriptional programs is not adequately addressed. Therefore, I do support publication at this time.

- The authors use several ovarian cancer cell lines in their experiments, but rationale for choice of cell line is not explicitly stated.

-A lot of key experimental details are missing from the manuscript. For example, I could not find any mention of what ovarian cancer line was used for the ChIPseq and RNAseq studies.

-Figure 3A needs a statistical test, such as hypergeometric enrichment, to support the conclusions raised.

- The major conclusion that MECOM/PRDM3 underlies PAX8-driven oncogenic functions in ovarian cancer is not fully supported by data presented. An experimental that demonstrates PRDM3 activity, such as a gain in histone methylation or some other feature associated with transcriptional regulation, at PAX8 recruited sites would better support this conclusion.
- The PAX8 and MECOM knockdown and ChIPseq experiments are nice additions to the manuscript, yet the data are not presented in a conventional manner. Importantly, it's unclear from the data how many significant changes were observed. I suggest using MA plots in Figure 3D.
- In regards to the ChIPseq and RNAseq experiments, little effort is made to connect Pax8 or PRDM3 binding to the gene expression changes observed following Pax8 or MECOM loss. If Pax8 is driving MECOM recruitment to specific loci, then one would expect any overlapping gene expression changes to be direct targets of this complex. These data would better support the idea that Pax8 and PRDM3 are regulating a specific gene expression program in ovarian cancer. Investigating those sites bound by both Pax8 and PRDM3 may shed some light into how this interaction governs specific transcriptional programs. Since there are subsets of Pax8 only and PRDM3 only bound sites, in addition to those sites bound by both factors, then you are likely to find something unique to the Pax8-PRDM3 interaction with further investigation.
- Only one de novo motif was identified in Figure 3B. Is this the only one? Where does it rank among other motifs? Was the C2H2 zinc finger motif in PRDM3 identified as well?
- The fluorescently tagged protein expression and FRAP Studies are underdeveloped and do not support the conclusion that Pax8 forms condensates. A reciprocal fluorescent tagging approach, such as Pax8-mCherry and PRDM3-GFP, is a proper control here and would rule out any issues related to reporter-protein aggregation, especially when it was concluded that Pax8-GFP forms molecular condensates. Also, these studies were conducted in U2OS osteosarcoma cells, not ovarian cancer cells.
- The sensitivity profiles shown in Figure 4A are not sufficiently described in the results section.
- The tumor growth studies in figure 4B,C need statistical tests.

Reviewer #3 (Remarks to the Author):

In their manuscript the authors describe the discovery and characterization of the interaction of PAX8 and PRDM3 (MDS1-EV11) and perform several experiments to shed light on functional aspects of this interaction and its role in cancer.

The experiments comprise Bio-ID mass spectrometry, Co-IP combined with Western Blot, Cross-link mass spectrometry, Nanobit complementation assays, NMR, FRAP, CRISPR tiling, RNA-seq and/or Chip-Seq in cell lines, patient derived xenografts and animal experiments.

The authors try to squeeze all these data into a single manuscript, an undertaking that makes it extremely difficult to follow the scientific line within the study and the manuscript. On one hand experimental data that does not necessarily contribute to the understanding of the protein function (at this stage) is provided while on the other hand essential experimental data is missing.

For instance,

1.) Missing data:

- Bio-ID MS: only raw data was deposited at PRIDE. This is not sufficient. Final results, intermediate files, experimental description and methods have to be submitted.
- Crosslink-MS: Only raw data was deposited at PRIDE. This is not sufficient. Final results, intermediate files, experimental description and methods have to be submitted.
- RNA-seq: data not submitted, no supplementary files with complete data
- Chip-Seq: data not submitted, no supplementary files with complete data (citation from manuscript: ..Raw data are currently being deposited to SRA..)

2.) FRAP: What is the conclusion of the higher mobility of PRDM3 in comparison to PAX8?

3.) NMR: The interaction has been shown already shown in Nanobit and Co-IP experiments. What is the additional value of this experiment? Which controls have been performed?

4.) Crosslinking - MS: Peptides are shown for each protein but which sites are crosslinked to each other?

5.) Chip-Seq: PAX8 and MECOM co-occupy only a subset of regions. According to the provided data MECOM only acts as a co-factor and does not bind to DNA. How do the authors differentiate clearly between PAX8-MECOM and PAX8-independent MECOM function? Which other genes are regulated by MECOM independent of PAX8?

6.) Discussion: The discussion is more a literature review on some function of PAX8 and MECOM than a (self-)critical discussion of the results in the context of the current knowledge. Clear conclusions are missing.

7.) The manuscript still contains some laboratory slang that should be corrected, e.g. 'the methyl region of PRDM3' - what is the methyl region of a protein?

8.) Similarly, on several occasions the authors use abbreviations that have not been introduced even in headlines,

e.g. ..PRDM3 (PR- + ZnF1-7).. — this abbreviation comes never up in the text

e.g. ..Paired (subdivided in PRD and Red).. - not found again in text or figure nor explained, different abbrev. in figures

9.) Xenograph models/Chip-Seq: I do not agree with the following conclusion:

These results argue for a dominant role for PAX8 in driving the gene expression program of ovarian cancer cells (yes) , by recruiting PRDM3/MECOM as a co-factor (no, where is the effect? Could be independent of PAX8).

A partial co-localization is detected, not the function. Similarly, the growth arrest of the xenograph models might be PAX8 independent.

From my point of view the major claims that ...PAX8-MECOM drive ovarian cancer... (title) and that the authors provide ..a detailed mechanistic characterization of the interaction of PAX8 and MECOM.. (first line discussion) are not fulfilled.

The author should provide results as supplementary data (e.g. tables) for review and also for the readers (MS, RNA-Seq, Chip-Seq). Minimal partial tables of selected results in figures are not sufficient.

Consequently, the current manuscript has to be considered too preliminary for publication.

Point-by-point response to the referees' comments

We thank the reviewers for the thorough and thoughtful assessment of our manuscript. All comments and suggestions have been considered and addressed, in large part by the inclusion of new experimental data and/or computational analyses as detailed in the point-by-point response below. Some of the newly provided information, which is intended to be for the reviewers' view only, has been included below each answer, but could be incorporated into the revised manuscript if deemed appropriate. We will be anyway happy to adhere to the transparent peer review process to allow these additional data/analyses to be accessible to the article readership. Overall, we thank the reviewers as the revisions that have stemmed from their comments have helped to improve the manuscript's clarity and further substantiate the original conclusions.

Reviewer #1 (Remarks to the Author):

This is a very comprehensive study of the role of PAX8 and MECOM in HGSOE. While PAX8 and MECOM have been implicated in HGSOE development for some time, this study demonstrates a physical interaction/dependency and provides insights into potential mechanisms of how this drives HGSOE. The authors have been very thorough with their experimentation by providing orthogonal validations where possible. A strength of the study is the use and validation in multiple ovarian cancer cell lines.

We would like to thank reviewer #1 for the positive comments and useful suggestions aimed at improving the quality of our manuscript. Please find below our point-by-point answers.

The manuscript would have been strengthened if in addition to knocking down PAX8 and MECOM expression in cancer cell lines, analyses could have been conducted in fallopian tumour epithelial, cell lines with induced expression. This could be an important experiment to understand the earliest events in HGSOE initiation since it seems PAX8 and MECOM are important early drivers of HGSOE.

We do acknowledge that understanding the sufficiency of PAX8 or MECOM in driving tumorigenesis could elucidate the disease origin etc. In our manuscript, we anyway focused on the dependency since PAX8 is a well-known, highly expressed marker of fallopian tube secretory epithelial cells (PMIDs: 17064757, 21317881 and many others). Additionally, a recent report has identified MECOM as bone fide 'master' regulator in HGSOE with lineage-restricted expression patterns in adult cancers (<https://www.biorxiv.org/content/10.1101/839142v1>). MECOM RNA and protein are also highly expressed in fallopian tube precursor cells (Reddy et al, manuscript under review). Indeed, in recently published scRNA-seq experiments (Hu et al., Cancer Cell, 2020, see *Figure 1 for reviewers* below), MECOM was significantly expressed in FTSECs. Given the pervasive high-level expression of PAX8 and MECOM in HGSOE precursor cells in the fallopian tube, this suggests that overexpression might not be the primary mechanism for promotion of oncogenesis, and that alternative mechanisms may be involved. This is in line with our previous work on PAX8 in RCC (Bleu et al., Nature Communications, 2019) and others' in ovarian cancer (Elias et al., JCI insight, 2016) showing that cistrome reshaping rather overexpression distinguish PAX8-driven cancers.

Elucidating the specific mechanisms underpinning altered PAX8 and MECOM function in tumors, for example, by proteomic profiling of post-translational modifications and context-specific binding partners, are long-term studies that are beyond the scope of the current report.

[REDACTED]

Figure 1 for reviewers

The manuscript could probably be shortened as parts of the results include text which is more appropriate for the discussion (e.g. the last paragraph of the results). Also the discussion summarises a lot of the biology of PAX8 from thyroid cancer but I think it would be useful to include some discussion of other work that has proposed targeting PAX8/MECOM. Specifically, how do the results of this study aid in developing new therapeutics?

We do acknowledge that the last part of the results sounded redundant with the first paragraph of the discussion, hence it was removed.

We additionally shortened our discussion, leaving space for further clarification of the critical relevance of this study reporting a previously uncharacterized interaction between PAX8 and PRDM3. While we have not genetically recapitulated the disruption between PAX8 and MECOM (see comments to reviewer #3), we believe that our biochemical / cellular / in vivo / human data strongly suggest that such interaction could be important for cancer.

It is then conceivable that, thanks to our report, assays aiming at measuring PAX8-PRDM3 binding could be designed in order to screen for molecules disrupting such interaction and probe its relevance in the cancer cell proliferation phenotype.

Overall the manuscript is well written, although there are a few sentences where the grammar is a bit awkward.

Results, page 3 "...its transcriptional network we utilized of the BioID system.

Results page 3 "...gene of interest allowing to label proximally"

Results, page 4 "while displaying similar protein expression levels confirmed (Figure S2A)".

Results page 6. "Importantly, lines highly sensitive PAX8 top were also highly sensitive to MECOM knockdown".

Discussion page 8. "PAX8 knockout (KO) mice die to the lack of formation of follicular".

Thank you for spotting these mistakes, we fixed them in the revised manuscript.

Reviewer #2 (Remarks to the Author):

Bleu et al. describe a new interaction between a proto-oncogene originating the MECOM locus (MDS-EVI or PRDM3) and the transcription factor, PAX8. The PAX8-PRDM3 complex regulates genes involved in adhesion and extracellular matrix deposition and controls tumor cell growth. They further show that PRDM3 is amplified in ovarian cancer, and PAX8-PRDM3 regulated genes stratify ovarian cancer cases with poor prognoses. The role of PAX8 and expression of MECOM gene products in ovarian cancer have been reported and MECOM is known to play roles in other cancers. Although the authors provide strong biochemical data supporting a PAX8-PRDM3 interaction and demonstrate its physiological relevance in ovarian cancer, the functional consequences of the interaction are not well described in the data presented. How PRDM3 mediates PAX8 oncogenic transcriptional programs is not adequately addressed. Therefore, I do support publication at this time.

We thank Reviewer #2 for the appreciation of our biochemical and cellular interaction data. Below, we attempted to address some of the concerns related to the transcriptional consequences of PAX8-PRDM3 binding/co-recruitment.

- The authors use several ovarian cancer cell lines in their experiments, but rationale for choice of cell line is not explicitly stated.

We employed 5 cell lines that were available to us among the ones showing the highest sensitivity to MECOM knockdown according to the RNAi sensitivity profile in DEPMAP reported in figure S4A and marked with blue asterisk in Figure 2 for reviewers below (OVTOKO was not available, RMGI is a PAX8 amplified model possibly skewing results). In particular, NIH:OVCA3 cell line was selected for ChIP-seq studies because of its amenability for in-vivo subcutaneous growth, to allow for in-vitro / in-vivo RNA-seq comparison and efficacy studies.

Figure 2 for reviewers

-A lot of key experimental details are missing from the manuscript. For example, I could not find any mention of what ovarian cancer line was used for the ChIPseq and RNAseq studies.

We used NIH:OVCA3 cells and added such information to the main text.

-Figure 3A needs a statistical test, such as hypergeometric enrichment, to support the conclusions raised.

We apologize for not including earlier a statistical test. Significance of overlaps between PAX8 and PRDM3 binding sites was assessed using a permutation test from R package ChIPpeakAnno (peakPermTest, version 3.24.1). We generated a random peak list based on TF Binding site clusters from Encode (wgEncodeTfbsV3) or gene-rich regions in Hg19 from UCSC to estimate the null distribution. The test, iterated 1000 times, demonstrated a statistically significant overlap between PAX8 and PRDM3 binding sites (p-value = 0.001) independently of the random peak list employed.

Using a random set of TFBS ENCODE clusters

P-value: 0.000999000999000999
Z-score: 151.4839
Number of iterations: 1000
Alternative: greater
Evaluation of the original region set: 4652
Evaluation function: cntOverlaps
Randomization function: randPeaks

Using a random set of regions in gene-rich loci

P-value: 0.000999000999000999
Z-score: 497.5688
Number of iterations: 1000
Alternative: greater
Evaluation of the original region set: 4652
Evaluation function: cntOverlaps
Randomization function: randPeaks

Figure 3 for reviewers

- The major conclusion that MECOM/PRDM3 underlies PAX8-driven oncogenic functions in ovarian cancer is not fully supported by data presented. An experimental that demonstrates PRDM3 activity, such as a gain in histone methylation or some other feature associated with transcriptional regulation, at PAX8 recruited sites would better support this conclusion.

PRDM3 has been reported to be an H3K9me1 methyltransferase in the cytoplasm marking de-novo histones for H3K9me3 labeling and heterochromatin formation (PMID: 22939622). We attempted to measure PRDM3 HMT activity or nucleotide binding by three methods: 1) Biochemical histone methyltransferase assay, 2) SAM binding by Scintillation Proximity assay (SPA) and 3) SAM or SAH binding by Differential Scanning Fluorimetry (DSF).

We failed to detect histone methyltransferase biochemical activity for the PR domain of PRDM3 (compared to the positive control DOT1L, either using an H3 tail peptide or purified nucleosomes) as well as PRDM3 is unable to bind SAM/SAH in SPA or DSF.

Biochemical histone methyltransferase assay

Scintillation proximity assay (SAM binding)

Differential scanning fluorimetry

Figure 4 for reviewers

In absence of a specific histone modification catalyzed by PRDM3 (we believe that the PR-domain contributes to the binding to PAX8 as previously reported for PRDM14-MTGR1 interaction, PMID: 26523391), we performed ChIP-seq for histone modifications associated to promoter activity (H3K4me3), transcriptional activity (H3K27ac) and gene repression (H3K27me3).

Data reveals that PAX8-PRDM3 co-occupied sites tend to be transcriptionally active enhancers and promoters as evidenced by high levels of H3K4me3 and H3K27ac and decreased H3K27me3, compared to PRDM3^{hi}-PAX8^{lo} sites and particularly PAX8 only sites (Panel A figure 5 for reviewers below).

In order to characterize the functional consequences of PAX8-dependent recruitment of PRDM3 on transcriptional regulation features, we segregated PAX8-PRDM3 co-bound sites based on LogFC of PRDM3 coverage upon PAX8 Knockdown (Panel B figure 5 for reviewers below). Interestingly, we

observed that sites in which PAX8 recruits PRDM3 (latter two bins) display decreased H3K27ac levels in cells bearing knockdown of PAX8 or knockdown of MECOM (albeit the latter at lower efficiency due to poorer knockdown as per main supplementary figure 3C) (Panel C figure 5 for reviewers below). These data are compatible with a model where PAX8/PRDM3 complex might recruited an acetyltransferase-containing complex, in line with the previously reported PAX8-p300¹ and MECOM-CBP/P/CAF² interactions. We then sought to analyze the transcriptional consequences of such decrease of H3K27ac at PAX8-dependent PRDM3-recruited sites (peaks with PRDM3 occupancy LogFC < -0.5) versus “PRDM3-unaffected sites”. We selected genes commonly regulated by PAX8 and PRDM3 (abs(FC) > 1.5) and performed pathway enrichment analysis. Interestingly “PRDM3-recruited sites” are assigned to genes involved in cell adhesion, signaling pathways and ECM (as reported for the multi-cell lines-derived signature in Figure 3F). Conversely, “PRDM3-unaffected genes” are enriched in other types of pathway categories (Panel D figure 5 for reviewers below). In summary, we believe that these data establish correlative relationship between PAX8-PRDM3 interaction on chromatin, histone acetylation and transcriptional activity.

Figure 5 for reviewers

-The PAX8 and MECOM knockdown and ChIPseq experiments are nice additions to the manuscript, yet the data are not presented in a conventional manner. Importantly, it's unclear from the data how many significant changes were observed. I suggest using MA plots in Figure 3D.

We thank the reviewer for the valuable suggestion. Accordingly, we generated MA plots of the data (see Figure 6 for reviewers below) and replaced figure 3D. Non-significant data points are represented as density (to avoid overplotting) while statistical significant differential binding is represented with a dot.

Figure 6 for reviewers

-In regards to the ChIPseq and RNAseq experiments, little effort is made to connect Pax8 or PRDM3 binding to the gene expression changes observed following Pax8 or MECOM loss. If Pax8 is driving MECOM recruitment to specific loci, then one would expect any overlapping gene expression changes to be direct targets of this complex. These data would better support the idea that Pax8 and PRDM3 are regulating a specific gene expression program in ovarian cancer. Investigating those sites bound by both Pax8 and PRDM3 may shed some light into how this interaction governs specific transcriptional programs.

We do agree that establishing a relationship between PAX8/PRDM3 genomic occupancy, their reciprocal recruitment and the consequent transcriptional response, would be an important information to support the notion that PAX8-PRDM3 complex drives a specific gene expression program. Indeed in our manuscript we focused on the common co-bound or co-regulated genes. In order to provide the complete datasets to the readership as well as a first analysis of the relationship between PAX8/MECOM binding and transcriptional consequences, we generated the following data/analyses:

- We provide the full list of PAX8/PRDM3 peaks in NIH:OVCAR3 cells (according to overlapping or divergent occupancy) in Supplementary table S3. We provide the full transcriptome LogFC data of the transcriptional changes observed by RNA-seq in NIH:OVCAR3 cells upon knockdown of PAX8 or PRDM3 in Supplementary table S4.
- Evaluated gene ontologies of the closest genes to PAX8-only, PRDM3-only or co-bound peaks (see Figure 7 for reviewers below).
- Evaluated the Pathway ontologies enriched in genes co-regulated by PAX8 and PRDM3 assigned to genomic sites in which PAX8 recruits PRDM3 (see Figure 5 for reviewers, panel D,

above). Of note the Pathway categories enriched resemble the ones enriched in the PAX8-MECOM signature (Figure S3E) identified by integrative analyses of the multi-cell-lines RNA-seq experiments.

Since there are subsets of Pax8 only and PRDM3 only bound sites, in addition to those sites bound by both factors, then you are likely to find something unique to the Pax8-PRDM3 interaction with further investigation.

As suggested by the reviewer, we investigated genes associated to PAX8-only bound, PRDM3-only bound or PAX8-PRDM3 co-bound sites using GREAT (great.stanford.edu). See below the Gene ontology analyses for Biological Processes and Molecular Functions for the closest genes associated to the indicated peaksets. We observed that PRDM3 sites (either alone or with PAX8) are both associated to genes involved in cell adhesion and extracellular matrix. However, the PRDM3 sites co-bound genes are enriched in developmental processes, while PRDM3-only sites are enriched in genes regulating MAPK cascade. PAX8-only sites are involved in stem cells and transcription.

It thereby appears that PRDM3, when co-bound with PAX8 drives a specific program aiming at regulating developmental/stemness genes involved in cell adhesion.

Figure 7 for reviewers

-Only one de novo motif was identified in Figure 3B. Is this the only one? Where does it rank among other motifs? Was the C2H2 zinc finger motif in PRDM3 identified as well?

We apologize for the misunderstanding. In figure 3B we only reported the top de novo motif identified by HOMER (<http://homer.ucsd.edu/homer/>). Here we show the top 10 de novo motifs identified in such analysis (Figure 8A for reviewers below)

The top motif identified is assigned to ZNF416 motif in the Jaspar database, however, this is due to the fact that ZNF416 motif (TGCCCAG) is significantly shorter (note the low information content bases next to the core motif). Indeed the second most similar motif in the Jaspar database is the one of PAX8 (look in the red shaded inlet in Figure 8A for reviewers below).

We observed enrichment of a motif resembling PRDM1 at 6th position. Notably all the motifs resembling such motifs contain the core motif AGA(A/G)A. While this motif resembles the previously identified C2H2 zinc finger motif in PRDM3 (PMID: 8321231), it was previously reported that the high affinity sites for PRDM3 include the core motif AGATA or AGACA.

Since Homer de novo motif finding is subjected to variability due to random sampling of background regions, we performed motif finding 10 times and reported how many times a given motif is retrieved among the top 10 motifs (Figure 8B for reviewers below). Of note, the ZNF416-PAX8 motif is identified 9/10 times, while the PRDM1 motif is identified only twice. Thereby we believe that co-bound sites are characterized predominantly by the presence of the PAX motif.

A

Rank	Motif	P-value	Best Match/Details
1		1e-464	ZNF416(Zf)/HEK293-ZNF416.GFP-ChIP-Seq(GSE58341)/Homer(0.720)
2		1e-457	CRZ1(MacIsaac)/Yeast(0.726)
3		1e-380	TEAD3/MA0808.1/Jaspar(0.944)
4		1e-358	AP-2alpha(AP2)/Hela-AP2alpha-ChIP-Seq(GSE31477)/Homer(0.778)
5		1e-355	RAV1(2)(AP2/EREBP)/Arabidopsis thaliana/AthaMap(0.647)
6		1e-351	PRDM1/MA0508.3/Jaspar(0.793)
7		1e-342	NFIX/MA0671.1/Jaspar(0.864)
8		1e-340	SeqBias: G/A bias(0.982)
9		1e-336	DoZ/MA0020.1/Jaspar(0.672)
10		1e-324	RAMOSA1/MA1416.1/Jaspar(0.800)

ZNF416(Zf)/HEK293-ZNF416.GFP-ChIP-Seq(GSE58341)/Homer

Match Rank: 1
Score: 0.72
Offset: 5
Orientation: reverse strand
Alignment: CTGSGARCCCAG---
-----TGCCCAGNH

Pax8(Paired/Homeobox)/Thyroid-Pax8-ChIP-Seq(GSE26938)/Homer

Match Rank: 2
Score: 0.63
Offset: -1
Orientation: reverse strand
Alignment: -CTGSGARCCCAG--
-SCAGYCADCCATGAC

Hand1:1/Tc5/MA0092.1/Jaspar

Match Rank: 3
Score: 0.57
Offset: 5
Orientation: reverse strand
Alignment: CTGSGARCCCAG---
-----ATCCCAGACN

PRDM1/MA0508.3/Jaspar

Match Rank: 1
Score: 0.79
Offset: 0
Orientation: reverse strand
Alignment: CARAGAAAGN-
-NAGAGAAAGNA

SRSF10(RRM)/Homo_sapiens-RNCMPT00019-PBM/HughesRNA

Match Rank: 2
Score: 0.78
Offset: 1
Orientation: forward strand
Alignment: CARAGAAAGN
-NAGAGAAAG-

Tb_0220(RRM)/Trypanosoma_brucei-RNCMPT00220-PBM/HughesRNA

Match Rank: 3
Score: 0.78
Offset: 2
Orientation: reverse strand
Alignment: CARAGAAAGN
--NAGAAAG-

B

Motif	# times motif is in top 10 rank across 10 runs
ZNF416 (PAX8)	9
CRZ1	6
SeqBias: G/A bias	6
TEAD2	5
SeqBias: GCW-triplet	3
MET28	2
ETV4	2
SMADs	2
POL009.1 DCE S II	2
PRDM1	2
POL0091.1 Zbtb3_1	2
SCL	2
Zfx	2
EBF1	2
TEAD	2

Figure 8 for reviewers

Within the co-bound sites, we additionally attempted to understand what are the features for the subsets of sites in which we observed changes in occupancy of PAX8 upon shMECOM or PRDM3 occupancy upon shPAX8 using MONALisa (<https://fmicompbio.github.io/monaLisa/articles/monaLisa.html>). MONALisa creates equal-size bins based on the logFC of occupancy upon perturbation (shades of orange to purple) and evaluates the enrichment of TF motifs in each bin (Figure 9 for reviewers below).

Interestingly we observed significant enrichment of PAX motifs in genomic loci in which the knockdown of MECOM decreases PAX8 occupancy indicating that a small subset of high affinity PAX8 sites require MECOM presence for PAX8 stability.

Conversely, for sites in which PRDM3 is recruited by PAX8, we have not observed enrichment of any TFs, indicating that the recruitment is likely dependent on an epigenetic mechanism, rather than sequence-specific.

Interestingly, we observed an enrichment of AP-1 motifs in PRDM3 sites that gain PRDM3 occupancy upon PAX8 knockdown, further suggesting that PAX8 might perturb PRDM3-AP1 occupancy. These data are intriguing in light of the data of Figure7 for reviewers showing that PRDM3-only sites associate to genes involved in MAPK signaling. (Figure 9 for reviewers below).

Additional analyses will focus on dissecting the different binding events observed to further elucidate how different pools of cellular PAX8 and or PRDM3 exert different effects on transcription.

*Figure 9 for reviewers*

-The fluorescently tagged protein expression and FRAP Studies are underdeveloped and do not support the conclusion that Pax8 forms condensates. A reciprocal fluorescent tagging approach, such as Pax8-mCherry and PRDM3-GFP, is a proper control here and would rule out any issues related to reporter-protein aggregation, especially when it was concluded that Pax8-GFP forms molecular condensates. Also, these studies were conducted in U2OS osteosarcoma cells, not ovarian cancer cells.

In order to address potential artefact due to the use of GFP as a fluorophore for confocal microscopy we performed the experiment in different conditions (Figure 10 for reviewers below):
- tagging of PAX8 or PRDM3 at either N- or C- terminus does not affect PAX8 or PRDM3 localization

- PAX8 condensation is dependent on its transactivation domain (as previously reported for intrinsically disordered regions, PMID: 30449618)
- we swapped the fluorophores used with monomeric mTurquoise2 and mVenus and we still observe PAX8 condensation

Figure 10 for reviewers

Moreover, we performed similar confocal fluorescence experiments in IGROV1 ovarian cancer cell line (the only cell line that we could efficiently transfect) and we observed a similar pattern of condensation for PAX8 (in a TA domain dependent manner) and homogeneous distribution of PRDM3 (Figure 11 for reviewers below). We additionally improved the quality of the plot of figure 3D and changed the colors to a colorblind-safe palette.

Figure 11 for reviewers

-The sensitivity profiles shown in Figure 4A are not sufficiently described in the results section.

We tried to use a clearer wording for explaining figure 4A in the main text.

-The tumor growth studies in figure 4B,C need statistical tests.

We performed post-hoc T-test on the mentioned experiments and observed statistical significant reduction of tumor growth upon PAX8 or MECOM knockdown with both shRNAs used. We updated figure 4B and 4C.

Reviewer #3 (Remarks to the Author):

In their manuscript the authors describe the discovery and characterization of the interaction of PAX8 and PRMD3 (MDS1-EVI1) and perform several experiments to shed light on functional aspects of this interaction and its role in cancer.

The experiments comprise Bio-ID mass spectrometry, Co-IP combined with Western Blot, Cross-link mass spectrometry, Nanobit complementation assays, NMR, FRAP, CRISPR tiling, RNA-seq and/or Chip-Seq in cell lines, patient derived xenographs and animal experiments.

The authors try to squeeze all these data into a single manuscript, an undertaking that makes it extremely difficult to follow the scientific line within the study and the manuscript. On one hand experimental data that does not necessarily contribute to the understanding of the protein function (at this stage) is provided while on the other hand essential experimental data is missing.

We thank the reviewer for acknowledging the large amount of multidisciplinary work reported in our manuscript and we apologize for the potential lack of clarity. We attempt to clarify the reported points below.

For instance,

1.) Missing data:

- Bio-ID MS: only raw data was deposited at PRIDE. This is not sufficient. Final results, intermediate files, experimental description and methods have to be submitted.
- Crosslink-MS: Only raw data was deposited at PRIDE. This is not sufficient. Final results, intermediate files, experimental description and methods have to be submitted.
- RNA-seq: data not submitted, no supplementary files with complete data
- Chip-Seq: data not submitted, no supplementary files with complete data (citation from manuscript: ..Raw data are currently being deposited to SRA..)

We apologize for the delays in depositing all the large-scale datasets due to technical challenges in the interface between our network and the repository network. We now provide complete raw datasets as well as processed data.

- *Bio-ID MS.* We now added a supplementary table in the manuscript including the proteins used for statistical analyses (removing non-unique peptides proteins) (Table S1). The PRIDE submission (PXD021709) includes all the raw data as well as the complete list of proteins/assigned spectra. We provide quite detailed material and methods in the manuscript, also considering that these assay is quite well established in the field (we performed standard data-dependent acquisition runs). Moreover, the expert reader can recover every detail of the method and instrument settings in the raw files using the appropriate Thermo software (e.g. using Xcalibur or Freestyle or with the MSFileReader or RawFileReader C++ libraries).
- *Crosslinking MS.* While the PRIDE submission (PXD021708) already contains an extensive Crosslink Spectral Match (CSM) table including all the requested info, we now include a simplified supplementary table in the manuscript (Table S2), focusing only on the observed intermolecular crosslinks, to facilitate access to our results to the broader community. The mass spec methodology follows standard manufacturer-recommended settings, which have also been extensively described in the literature, e.g. by the Heck lab (Liu F et al., Nat Methods. 2015 (12):1179; Liu F et al., ref. 38) and in a summarized form by the authors in ref. 37. Moreover, the expert reader can recover the detail of the method and instrument settings in the raw files using the appropriate Thermo software such as Xcalibur or Freestyle or with the MSFileReader or RawFileReader C++ libraries.
- *RNA-seq.* We apologize for not providing earlier the SRA accession details. Data can now be found on SRA <https://www.ncbi.nlm.nih.gov/sra> with Identifiers SRP276625 and BioProject PRJNA655836. We additionally provide a multitable excel file as supplementary table (Table S4) including both filtered and unfiltered differential expression values for all the experimental conditions reported in the manuscript.
- *ChIP-Seq.* Similarly to the RNA-seq data, Raw data can now be accessed at SRA using Identifiers SRP276627 and BioProject PRJNA655844. We now also include a multitable excel file (Table S3) with the all the processed data including identified peaks in every condition and differential binding analyses.

2.) FRAP: What is the conclusion of the higher mobility of PRDM3 in comparison to PAX8?

We do believe that the FRAP data reveal that PAX8 can be strongly tethered to hub-like structures while PRDM3 higher mobility is suggestive of a role as a co-factor (we added this sentence to the manuscript). We are currently focusing on single particle tracking experiments in order to evaluate the single molecule

spatial dynamics of the interactions between PAX8 and PRDM3 in such PAX8 hub-like structures as well as to describe the nature of such structures (transcriptional hubs?). Such investigations will be part of a future manuscript.

3.) NMR: The interaction has been shown already show in Nanobit and Co-IP experiments. What is the additional value of this experiment? Which controls have been performed?

NanoBit and Co-IP experiments aimed at demonstrating that PAX8 and MECOM resided in the same complex, but failed to demonstrate a binary physical interaction. We believe our NMR studies demonstrated that the two purified proteins are able to interact. Additionally we demonstrated that those two specific construct boundaries are sufficient for the interaction.

We performed additional controls by demonstrating that the PAX8 DNA binding domain protein is competent for DNA binding to an oligo containing its consensus motif.

Figure 12 for reviewers

We performed an additional negative control to our NMR experiments by analyzing chemical shifts of labeled PAX8 DNA binding domain when mixed with purified PRDM3 PR-/SET- domain (75-200 boundary).

In line with the NanoBit data on Figure 2E, both the PR-/SET- domain as well as the first array of ZnF domains of PRDM3 are necessary for optimal interaction. This is evidenced by the lack of substantial chemical shifts in the orange and green boxes and general lack of peak broadening in the right plot below.

2D NMR of PAX8 DNA binding domain upon addition of PRDM3 constructs

Figure 13 for reviewers

4.) Crosslinking - MS: Peptides are shown for each protein but which sites are crosslinked to each other?

We apologize for the lack of clarity. We modified Figure S2G with a table that clarifies which peptides and which residues are involved in the intermolecular crosslinks.

5.) Chip-Seq: PAX8 and MECOM co-occupy only a subset of regions. According to the provided data MECOM only acts as a co-factor and does not bind to DNA. How do the authors differentiate clearly between PAX8-MECOM and PAX8-independent MECOM function? Which other genes are regulated by MECOM independent of PAX8?

We thank the reviewer for the important comment. By ChIP-seq indeed we defined sites that are PAX8-only, PRDM3-only and co-bound sites. Of note, even in PRDM3-only sites, there is residual signal for PAX8 (see heatmap in figure 5A for reviewers). Nevertheless we interrogated the gene ontology of genes associated with the three genomic categories and, while PRDM3 occupancy (either with or without PAX8 presence) is associated to genes involved in cell adhesion, PRDM3-only sites seem to be associated to genes involved in MAPK signaling while co-bound sites are associated to developmental genes (Figure 7 for reviewers above). Such distinction is intriguing since, even when focusing on co-bound sites, PRDM3 sites that gain PRDM3 occupancy upon PAX8 knockdown are enriched in DNA binding motifs of AP-1 transcription factors (downstream effectors of MAPK signaling) as evidenced in figure 9 for reviewers above.

Specifically to the question of which genes are regulated by MECOM. We performed integrative analyses of the RNA-seq dataset across multiple cell lines, filtering based on consistent regulation by shMECOM in >3 cell lines and filtering out genes regulated by PAX8 in any cell line.

Based on this, we identified a signature of 47 genes (see heatmap in the Figure 14A for reviewers below). Several proteins involved in cell adhesion and extracellular matrix are readily evident (e.g. HBEGF, FN1, ITGB5 etc. etc.) and future studies will focus on understanding the specific role of MECOM beyond PAX8 engagement.

Figure 14 for reviewers

6.) Discussion: The discussion is more a literature review on some function of PAX8 and MECOM than a (self-)critical discussion of the results in the context of the current knowledge. Clear conclusions are missing.

Thanks for the comment aiming at improving our Discussion (as noted by reviewer #1). We significantly shrunk the portion of the discussion which extensively described the role of PAX8 or PRDM3 during development and tried to refocus on our findings and the potential implication from the therapeutic perspective.

7.) The manuscript still contains some laboratory slang that should be corrected, e.g. 'the methyl region of PRDM3' - what is the methyl region of a protein?

The sentence "the methyl region of PRDM3" should read "The methyl region of the proton spectrum of PRDM3". We corrected it in the revised manuscript.

8.) Similarly, on several occasions the authors use abbreviations that have not been introduced even in headlines,
e.g. ..PRDM3 (PR- + ZnF1-7).. — this abbreviation comes never up in the text
e.g. ..Paired (subdivided in PRD and Red).. - not found again in text or figure nor explained, different abbrev. in figures

We apologize for these abbreviations, which might have created confusion. We fixed them in the main text.

9.) Xenograph models/Chip-Seq: I do not agree with the following conclusion:

These results argue for a dominant role for PAX8 in driving the gene expression program of ovarian cancer cells (yes) , by recruiting PRDM3/MECOM as a co-factor (no, where is the effect? Could be independent of PAX8).

A partial co-localization is detected, not the function. Similarly, the growth arrest of the xenograph models might be PAX8 independent.

From my point of view the major claims that ...PAX8-MECOM drive ovarian cancer... (title) and that the authors provide ..a detailed mechanistic characterization of the interaction of PAX8 and MECOM.. (first line discussion) are not fulfilled.

We thank the reviewer for the comment and agree that sentences stating that PAX8 oncogenic functions depend on PRDM3/MECOM interaction might overstate our conclusion. For this reason, we decided to tone down our conclusions in those two specific instances.

Indeed a proper experimental setup to prove the relative contribution of PRDM3 interaction to PAX8 function would require the identification of a PAX8 mutant unable to interact with PRDM3, while retaining other functions.

We made attempts to identify such mutant by setting up a biochemical assay by NanoBit with purified recombinant proteins (panel A below). Such assay revealed an interaction affinity between PAX8 (9-135) and PRDM3 (75-434) around 1 μ M and such interaction is largely retained by a DNA-binding mutant of PAX8 (C45E-C57E) (panel B and C below).

We then performed an alanine scanning for the DNA binding domain of PAX8 by high-throughput cloning and expression of a library of alanine mutants of PAX8 (9-135) LgBit constructs. Protein expression in each well was measured upon addition of the HiBit high-affinity LgBit ligand and protein interaction with PRDM3 was measured by adding PRDM3 75-434-SmBit protein (panel D below).

While a subset of PAX8 mutants displayed low expression/solubility, we observed that mutants around aminoacids 118-125 retained high level of expression (x-axis in panel E) while displaying decreased PRDM3 interaction (y-axis in panel E).

Independent validation from large-scale purification of PAX8 (9-135) mutants confirmed that the region 118-125 contributes to the interaction with PRDM3 (panel F). However, none of the mutants was able to decrease the interaction by more than 50% compared to PAX8 WT protein or negative control mutants. (panel F). Importantly, the identified interacting region resides in an area close to two peptides that contained lysines labeled in our CX-MS experiment (panel G and main figure 2H) further corroborating the validity of our interaction mapping strategy.

Since the identified region is known to contribute to DNA interaction, we believe that more dramatic mutations in such region might not only completely disrupt PAX8-PRDM3 but also PAX8-DNA interaction.

Thereby further studies are needed to conclusively disentangle the relative contribution of PAX8-PRDM3 interaction to PAX8 oncogenic functions.

Figure 15 for reviewers

Nevertheless, we believe that both PAX8 and PRDM3/MECOM are drivers of the disease, as stated in the title, based on the significant phenotype observed in vitro and in vivo upon knockdown of either PAX8 or MECOM (figure 4A – 4C). Additionally the dramatic loss of both PRDM3 chromatin occupancy (figure 3D) and total protein levels of MECOM isoforms (figure 4D) observed upon PAX8 knockdown are highly suggestive of a potential contribution of MECOM binding to PAX8 oncogenic function. Such hypothesis is further corroborated by our correlation findings in large scale functional genomics and transcriptomics datasets (Figure 4A, 4E, S4A, S4D, S4E).

The author should provide results as supplementary data (e.g. tables) for review and also for the readers (MS, RNA-Seq, Chip-Seq). Minimal partial tables of selected results in figures are not sufficient.

As mentioned in point 1, we now include supplementary tables for the following datasets:

- Table S1: BioID-MS
- Table S2: Crosslinking-MS
- Table S3: ChIP-seq datasets
- Table S4: RNA-seq datasets

REVIEWERS' COMMENTS

Reviewer #1 (Remarks to the Author):

The authors have provided a very detailed and considered rebuttal of the comments and suggestions of the reviewers. Sufficient changes have been made to the manuscript where necessary and where changes have not been made the authors have provided good reasoning to support the data as presented.

Reviewer #2 (Remarks to the Author):

I have no further comments.

Reviewer #3 (Remarks to the Author):

The authors have spent great effort on the reviewers comments and they have, from my point of view, answered all issues raised by the reviewers.

However, I have to insist on a minor but essential technical aspect.

The MS data submission has to be complete.

Some properties of the datasets are missing.

These are essential keywords for search and retrieval:

Experiment Properties:

Submission PXD021709

- 1) Experiment type (BioID Proximity Labeling)
- 2) Organism part (here: cell culture, IGROV-1-PAX8-BioID-T2A-mCherry)
- 3) Software: (here: Proteome Discoverer V2.1)

Submission PXD021708

- 1) Organism ..
- 2) Correction required for Bruker Ultraflex extreme II
- 3) Software(s) ..
- 4) Experiment type(s) ..
- 5) Quantification ..

Point-by-point response to the referees' comments (II)

We thank you very much the reviewers for the careful and insightful comments which lead to the improvement and strengthening of our manuscript via the revision process. See below the response to the specific point by Reviewer #3.

Reviewer #1 (Remarks to the Author):

The authors have provided a very detailed and considered rebuttal of the comments and suggestions of the reviewers. Sufficient changes have been made to the manuscript where necessary and where changes have not been made the authors have provided good reasoning to support the data as presented.

Reviewer #2 (Remarks to the Author):

I have no further comments.

Reviewer #3 (Remarks to the Author):

The authors have spent great effort on the reviewers comments and they have, from my point of view, answered all issues raised by the reviewers.

However, I have to insist on a minor but essential technical aspect.

The MS data submission has to be complete.

Some properties of the datasets are missing.

These are essential keywords for search and retrieval:

Experiment Properties:

Submission PXD021709

- 1) Experiment type (BioID Proximity Labeling)
- 2) Organism part (here: cell culture, IGROV-1-PAX8-BioID-T2A-mCherry)
- 3) Software: (here: Proteome Discoverer V2.1)

Submission PXD021708

- 1) Organism ..
- 2) Correction required for Bruker Ultraflexreme II
- 3) Software(s) ..
- 4) Experiment type(s) ..
- 5) Quantification ..

We thank the reviewer to point out the missing details of the PRIDE submissions. We updated the submissions completing the indicated fields in accordance with the PRIDE controlled vocabulary. The submissions are now freely accessible to the public.